# Learning to Watermark in the Latent Space of Generative Models

**Sylvestre-Alvise Rebuffi**[1][*]  **Tuan Tran**[1][*]  **Valeriu Lacatusu**[1][*]  **Pierre Fernandez**[1]  **Tomáš Souček**[1]
**Nikola Jovanović**[2]  **Tom Sander**[1]  **Hady Elsahar**[1]  **Alexandre Mourachko**[1]

## Abstract

Existing approaches for watermarking AI-generated images often rely on post-hoc methods applied in pixel space, introducing computational overhead and visual artifacts. In this work, we explore latent space watermarking and introduce DISTSEAL, a unified approach for latent watermarking that works across both diffusion and autoregressive models. Our approach works by training post-hoc watermarking models in the latent space of generative models. We demonstrate that these latent watermarkers can be effectively distilled either into the generative model itself or into the latent decoder, enabling in-model watermarking. The resulting latent watermarks achieve competitive robustness while offering similar imperceptibility and up to $20\times$ speedup compared to pixel-space baselines. Our experiments further reveal that distilling latent watermarkers outperforms distilling pixel-space ones, providing a solution that is both more efficient and more robust. Code and model are available at https://github.com/facebookresearch/distseal.

## 1. Introduction

The rapid advancement of generative models has enabled the creation of increasingly realistic synthetic content, raising serious concerns about potential misuse, including the generation of harmful content, deepfakes, and intellectual property violations (Chesney & Citron, 2019). As outputs of these models become more indistinguishable from authentic content, the need for reliable provenance mechanisms grows. Invisible watermarking offers a promising solution by embedding imperceptible signals into generated content that can later verify its synthetic origin, enabling accountability while preserving the utility and quality of generated outputs.

Recent watermarking research can be categorized into *post-hoc* methods that modify already-generated content in pixel-space (Bui et al., 2024; Fernandez et al., 2024), *out-of-model* generation-time methods that alter the sampling process (Wen et al., 2023; Yang et al., 2024; Jovanović et al., 2025; Lukovnikov et al., 2025), and *in-model* methods where watermarks are directly integrated into the generative model weights (Yu et al., 2022; Fernandez et al., 2023; Kim et al., 2024).

While post-hoc pixel-space watermarking has emerged as the method of choice for industry deployment (Gowal et al., 2025; Castro, 2025) due to its flexibility and model-agnostic nature, it faces fundamental limitations. First, it incurs substantial computational overhead and latency as it operates on high-resolution pixel representations (e.g., $512\times512\times3$). Second, it provides weak to no security for open-source deployments, as users can trivially bypass watermarking by removing a single line of code. To this end, we explore whether it is possible to directly embed the watermarking into the generative models in such a way that (a) the watermarking does not introduce a latency penalty, and (b) it cannot be disabled even in the case of open-source models. Unlike the work of Fernandez et al. (2023), which adds watermarking capability into the Stable Diffusion decoder, we strive to watermark the latent outputs of a diffusion or autoregressive model directly.

In this work, we introduce **DISTSEAL**, a unified framework for latent watermarking that operates across both diffusion and autoregressive generative models. DISTSEAL works by adapting post-hoc watermarking methods for latent space, allowing us to watermark both continuous latent representations of diffusion models and discrete token sequences of autoregressive models. We show that these latent watermarkers can be distilled directly into the generative model itself, while retaining the robustness characteristics of the teacher watermarkers. Not only that, we demonstrate that latent watermarks are easier to distill than their pixel-space counterparts. In detail, our contributions are:

- We introduce DISTSEAL—a unified latent watermarking framework for both diffusion and autoregressive models. We show that DISTSEAL delivers up to

---

[*]Equal contribution  [1]Meta FAIR  [2]ETH Zurich. Correspondence to: Sylvestre-Alvise REBUFFI <sylvestre@meta.com>.

*Proceedings of the 43$^{rd}$ International Conference on Machine Learning*, Seoul, South Korea. PMLR 306, 2026. Copyright 2026 by the author(s).

20× speedup compared to pixel-space methods while achieving competitive robustness.

- We demonstrate that latent watermarkers can be effectively distilled into model weights (generative model or decoder). Critically, we show that distilling latent watermarkers is more effective than distilling pixel-space ones, allowing us to achieve state-of-the-art in-model watermarking for open-source generative models.

- We conduct a comprehensive study of post-hoc and in-model watermarking trade-offs, multi-watermarking compatibility, and watermark forgetting, providing actionable guidance for real-world deployment.

## 2. Related Work

**Image generative models.** Modern image generation relies on several paradigms, with diffusion models and autoregressive models being the most prominent. Diffusion models (Ho et al., 2020; Song et al., 2020; Dhariwal & Nichol, 2021) learn to reverse a noise diffusion process, with Latent Diffusion Models (LDMs) (Rombach et al., 2022) like Stable Diffusion operating in a compressed latent space via a Variational Autoencoder (VAE) (Kingma & Welling, 2013). Some recent works improve compression autoencoders for efficient high-resolution generation (Chen et al., 2024a; Xie et al., 2025). Autoregressive models (Esser et al., 2021; Ramesh et al., 2021; Yu et al., 2024) predict image tokens sequentially, with recent work including RAR-XL (Yu et al., 2025) and MaskBit (Weber et al., 2024). This approach notably allows for interleaved multimodal generation (Chameleon Team, 2024; Zhan et al., 2024; Wu et al., 2025a). Our work focuses on watermarking both diffusion and autoregressive models.

**Post-hoc image watermarking.** Traditional image watermarking methods operate in either the spatial domain by directly modifying pixels (Van Schyndel et al., 1994; Bas et al., 2002), or the frequency domain by embedding watermarks in transform coefficients such as DFT, DCT, DWT, etc. (Cox et al., 1997; Barni et al., 1998; Xia et al., 1998). Deep learning-based post-hoc watermarking methods have emerged as robust alternatives, typically employing encoder-decoder architectures trained end-to-end to embed imperceptible watermarks while maintaining robustness to various transformations (Zhu et al., 2018; Zhang et al., 2019; 2020; Yu, 2020; Luo et al., 2020; Ma et al., 2022; Jia et al., 2021). They are the method of choice for the industry (Bui et al., 2024; Xu et al., 2025; Gowal et al., 2025; Sander et al., 2025; Fernandez et al., 2024) due to their flexibility and ease of deployment. Our work adapts current post-hoc watermarking schemes for watermarking the latent space of generative models, offering computational efficiency while maintaining competitive robustness.

**Generation-time image watermarking.** Generation-time watermarking methods embed watermarks directly during content generation, eliminating post-processing overhead. RoSteals (Bui et al., 2023) is one such approach, and operates in the latent space of autoencoders using a coverless watermarking approach. For diffusion models, Stable Signature (Fernandez et al., 2023) performed in-model watermarking by fine-tuning the latent decoder, followed by extensions using hypernetworks (Kim et al., 2024) and adapters (Ci et al., 2025; Rezaei et al., 2024). Out-of-model approaches like Tree-Ring (Wen et al., 2023) and its variants (Hong et al., 2024; Ci et al., 2024; Lei et al., 2024) embed patterns in the initial noise of diffusion models or guide the generation process to produce watermarked outputs (Gesny et al., 2026). For autoregressive image models, several recent methods (Jovanović et al., 2025; Lukovnikov et al., 2025; Tong et al., 2025; Wu et al., 2025b; Hui et al., 2025; Meintz et al., 2025) address the challenge of watermarking discrete token sequences, each proposing different solutions to achieve reverse cycle consistency or robust embedding in the generation process. More related to our work of embedding watermarks directly into the generative models is AquaLoRA (Feng et al., 2024), which fine-tunes the diffusion U-Net with LoRA for coverless latent watermarking. In contrast, DISTSEAL is conditioned on the generated image; therefore, it can hide the watermark more effectively, ensuring its state-of-the-art robustness and imperceptibility.

## 3. Background and Problem Statement

### 3.1. Generative Models with Latent Representations

Modern generative models rely on autoencoders to compress images into a latent space. Let $x \in \mathbb{R}^{H \times W \times 3}$ denote an image, and $\mathcal{E}, \mathcal{D}$ denote the encoder and decoder. The encoder maps the image to $z = \mathcal{E}(x) \in \mathbb{R}^{h \times w \times c}$ where $h < H$, $w < W$, and the decoder reconstructs $\hat{x} = \mathcal{D}(z)$. Latent diffusion models (Rombach et al., 2022) iteratively denoise a latent representation $z_T$. The denoising network $G_\phi$ predicts noise or clean latents at each timestep $t$, producing the final latent $z_0$, which is decoded as $x = \mathcal{D}(z_0)$. Autoregressive models (Esser et al., 2021) quantize continuous latents into discrete tokens via $Q : \mathbb{R}^{h \times w \times c} \to \{1, \dots, K\}^{h \times w}$. The model $G_\phi$ predicts token sequences $\mathbf{s} = (s_1, \dots, s_N)$ autoregressively, which are then converted back to continuous latents and decoded to images.

### 3.2. Problem Statement

We aim to develop a unified method that addresses the two complementary tasks outlined below.

**Task 1: Post-hoc latent watermarking.** The post-hoc latent watermarker embeds an imperceptible watermark encoding a binary message $m \in \{0,1\}^K$ into the latent repre-

sentation before decoding, making it robust to transformations such as compression or geometric changes. It can be deployed at inference time with different binary messages, for instance, to distinguish between different generative models trained on the same autoencoder, or to detect AI-generated content. It operates similarly to post-hoc methods in pixel space, but offers faster inference by operating in a spatially compressed space.

**Task 2: In-model watermarking.** It modifies the generative model ($G_\phi$) or decoder ($\mathcal{D}$) to inherently embed a fixed watermark message during generation, eliminating the need for post-processing. This makes it a better option for open-source model releases where the watermark needs to be integrated directly into the model weights.

## 4. Method

DISTSEAL consists of two main training stages, illustrated in Fig. 1. First, we train a post-hoc watermark embedder-extractor pair (we refer to this pair as "watermarker"), where the embedder operates in the latent space of the generative model (Section 4.1). Second, we optionally distill this post-hoc watermarker into either the generative model itself or its latent decoder for in-model watermarking (Section 4.2).

### 4.1. Latent Watermarking

We consider a watermark embedder $W_\theta$ that takes as input the latent representation $z$ and a $K$-bit binary message $m$ and outputs a watermarked latent representation $z_w = z + \epsilon W_\theta(z, m)$ where $\epsilon$ is the scaling factor which controls the strength of the watermark. The watermarked image is then obtained as $x_w = \mathcal{D}(z_w)$. In comparison, a common post-hoc watermarking embedder directly modifies the pixels of $x$ to produce a watermarked image $x_w = x + \epsilon W_\theta(x, m)$.

For autoregressive models with discrete latent spaces, we apply the post-hoc embedder before quantization, obtaining a new sequence of discete tokens with $z_w = Q(z + \epsilon W_\theta(z, m))$ where $Q$ is the quantization function. We use straight-through estimation (Bengio et al., 2013) to back-propagate through the quantization step during training. We can also apply the embedder after quantization, meaning that we modify the embeddings of the discrete tokens rather than creating a new sequence of tokens.

Then, the watermarked images are augmented with various valuemetric, geometric and compression augmentations to simulate plausible edits done by users. Finally, the augmented images are fed to the watermark extractor $E_\theta$ which predicts the watermark message.

We train the watermark embedder $W_\theta$ and watermark extractor $E_\theta$ using a combination of watermark extraction loss and discriminator loss. The watermark extraction loss is a binary cross-entropy (BCE) loss between the predicted message bits and the ground-truth message $m$. The discriminator loss is a standard adversarial hinge loss (Weber et al., 2024), applied in the pixel space, tries to enforce that the watermarked image $x_w$ is indistinguishable from real images. For the quantized latents, we observe that the perturbations in the latent space introduced by the embedder can result in semantic modifications to the image. Therefore, we do not use reconstruction losses or perceptual losses for any of the post-hoc watermark embedder and only constrain the watermark with the watermarking strength $\epsilon$ and the discriminator loss. The overall loss function is given by:

$$\mathcal{L} = \lambda_w \mathcal{L}_w(x_w, m) + \lambda_{\text{disc}} \mathcal{L}_{\text{disc}}(x, x_w). \quad (1)$$

### 4.2. Distillation for In-Model Watermarking

We may optionally distill the latent watermarker into the generative model by watermarking the latents of the training images and fine-tuning on this watermarked data. Formally, given latents $z$ and a fixed message $m$ used across training images, we generate continuous or discrete watermarked latents $z_w = z + \epsilon W_\theta(z, m)$ or $z_w = Q(z + \epsilon W_\theta(z, m))$ using the trained embedder $W_\theta$. The generative model is then fine-tuned to reconstruct or predict $z_w$ instead of $z$,

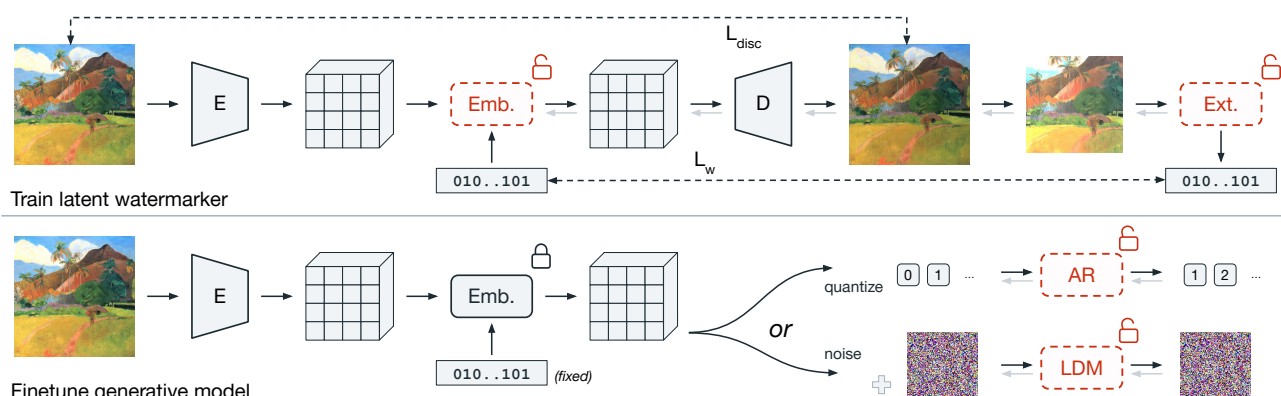

*Figure 1.* Overview of DISTSEAL. We train a post-hoc embedder/extractor pair, where the embedder operates in the latent space (top). We may then distill the embedder into the diffusion or autoregressive model (bottom). Fig. 2 details the distillation in the latent decoder.

encouraging it to produce watermarked latents during the generation process. The loss function for distillation in the generative model reads:

$$\mathcal{L}_{\text{gen}} = \mathcal{L}_{\text{recon}}(G_\phi(x), z_w), \qquad (2)$$

where $G_\phi$ denotes the generative model (diffusion or autoregressive), $\mathcal{L}_{\text{recon}}$ is the reconstruction loss (e.g., MSE or cross-entropy, resp.), and $x$ represents the input (noisy latent or token sequence, resp.). After distillation, the generative model embeds the watermark during the generation phase, eliminating the need for a separate post-hoc step.

This approach can also be applied to the latent decoder $\mathcal{D}$ by training it to reconstruct watermarked images from non-watermarked latents, i.e., by training $\mathcal{D}(z)$ to be close to $\mathcal{D}_\text{o}(z_w)$, where $\mathcal{D}_\text{o}$ is the original decoder and $z_w$ the watermarked latent. We optimize a combination of a reconstruction loss $\mathcal{L}_{rec}$ and a watermark extraction loss $\mathcal{L}_w$, which drives the extractor $E_\theta$ to recover the message $m$ from the distilled decoder's output:

$$\mathcal{L}_{\text{dist}} = \mathcal{L}_{rec}\left(\mathcal{D}_\text{o}(z_w), \mathcal{D}(z)\right) + \lambda_w \mathcal{L}_w\left(E_\theta(\mathcal{D}(z)), m\right). \qquad (3)$$

where $\mathcal{L}_{rec} = \ell_1 + \lambda_p \mathcal{L}_{\text{LPIPS}}$ combines pixel-wise $\ell_1$ loss and perceptual LPIPS loss (Zhang et al., 2018). Unlike Stable Signature, we train the decoder to reconstruct the watermarked image rather than just leveraging the signal from the extractor $E_\theta$ to guide the distillation.

## 5. Experiments

We first detail our experimental setup in Section 5.1, then we present our main results on post-hoc latent watermarking (Section 5.2) and in-model watermarking via distillation in the latent decoder (Section 5.3.1) and in the generative model (Section 5.3.2). Additional analysis on multi-watermarking, watermark formation, and watermark forgetting are provided in Appendices G, H and K.

### 5.1. Experimental Setup

We detail the experimental setup for our main experiments, and refer to Appendix A for additional details.

**Generative models.** For the diffusion model, we use the class-conditional ImageNet UViT-H model from

DCAE (Chen et al., 2024b) generating $8\times8\times128$ latents, which are decoded into $512\times512$ images. For the autoregressive model, we use RAR-XL (Yu et al., 2025), which is based on the MaskGIT-VQGAN (Chang et al., 2022) autoencoder that compresses $256\times256$ images into 256 discrete tokens with a codebook size of 1024. We show additional results for MaskBit (Weber et al., 2024) in Appendix I.

**Watermarking models.** For the pixel watermarking baseline, we utilize the default architecture from VideoSeal, which features a UNet embedder and a ConvNeXt-tiny extractor. For the latent watermarkers, we modify the embedder to accommodate the lower spatial resolution of the latent representations by removing all downsampling and upsampling layers while preserving the middle ResNet blocks. The extractor architecture remains unchanged across both methods. To perform a like-for-like comparison between pixel-space and latent-space watermarking, both approaches share identical training objectives, training schedules, augmentations, and hyperparameters, with only the watermark strength $\epsilon$ adjusted to account for the different value ranges between pixel and latent spaces. We do not use any JND masking for the pixel watermarker. We train all the models with 64-bit messages for 600k steps on ImageNet (Deng et al., 2009) with a batch size of 128. For all the models, the watermark strength is set to a higher value for the first 100k steps to kickstart training, and then it is decreased to its final value over the next 100k steps with a cosine decay. We provide additional training details in Appendix A.

**Distillation.** For the distillation in the diffusion model, we simply resume the training of the pre-trained transformer as in the original DCAE paper (Chen et al., 2024b) but using watermarked latents instead of the original latents for 100k steps. For the distillation in the autoregressive model, we resume the training of the pre-trained transformer as in the original RAR-XL paper (Yu et al., 2025) for 10k steps using watermarked token sequences instead of the original ones. For distillation in the latent decoder, we fine-tune the pre-trained decoder using the loss from Eq. 3 with Adam for 10k steps with batch size 16 and a learning rate ramping up for 1k steps to 1e-4 before a cosine decay. We use $\lambda_p = 0$ for DCAE and $\lambda_p = 1$ for RAR-XL ($\lambda_p = 0$ leads to blurry reconstructions for RAR-XL, see Figure 15).

**Metrics.** To evaluate watermarking robustness, we measure the bit accuracy of the extracted watermark after applying various transformations to the watermarked images. For post-hoc watermarking methods, we evaluate them on 50k images generated by the respective generative models. For in-model watermarking, we directly generate 50k images from the distilled generative models. To assess visual quality, we use (when available) the PSNR between original and watermarked images, which is better at assessing pixel-level

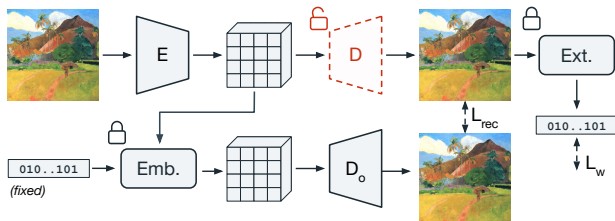

*Figure 2.* The decoder of the generative model can be distilled to produce watermarked images from non-watermarked latents.

*Table 1.* Comparing post-hoc watermarking methods on DCAE-generated images in terms of image quality and bit prediction robustness to various attacks. For post-hoc latent watermarking, we also report a coverless variant that is independent of the input.

| Method | PSNR | FID | IS | Identity | Valuemetric | Geometric | Compression | Combined | Avg |
|---|---|---|---|---|---|---|---|---|---|
| Post hoc pixel | 43.48 | 10.84 | 99.53 | 100.00 | 99.89 | 94.70 | 99.90 | 97.29 | 97.78 |
| Post hoc latent | 31.06 | 11.42 | 98.11 | 99.75 | 98.08 | 91.62 | 99.23 | 84.28 | 95.18 |
| Post hoc latent (coverless) | 30.90 | 11.29 | 99.02 | 97.69 | 95.85 | 87.25 | 96.74 | 73.94 | 91.81 |

*Table 2.* Comparing post-hoc methods on RAR-generated images on image quality and bit prediction after various attacks. For post-hoc latent watermarking, we compare the watermarking results before and after the quantization step ('before quant' and 'after quant').

| Method | PSNR | FID | IS | Identity | Valuemetric | Geometric | Compression | Combined | Avg |
|---|---|---|---|---|---|---|---|---|---|
| Post hoc pixel | 41.80 | 3.09 | 292.75 | 100 | 99.69 | 92.97 | 99.74 | 93.93 | 97.27 |
| Post hoc latent (before quant) | N/A | 3.56 | 239.00 | 96.92 | 95.91 | 87.20 | 95.98 | 77.00 | 90.60 |
| Post hoc latent (after quant) | 24.54 | 3.44 | 250.50 | 99.52 | 98.56 | 91.37 | 97.97 | 82.35 | 93.96 |

distortions, and FID (Heusel et al., 2017) and IS (Salimans et al., 2016), which are more suitable for evaluating perceptual quality and diversity of generated images. For FID and IS, we generate 50,000 images from the generative models (with and without watermarking) and compare them to the ImageNet validation set. We use classifier guidance for RAR-XL as in their paper, and no guidance for DCAE.

### 5.2. Post-Hoc Latent Watermarking Results

Here, we compare post-hoc latent watermarking with post-hoc pixel watermarking on images outputted by the DCAE and RAR-XL generative models.

**Continuous latent space.** Table 1 shows results for the DCAE diffusion model. The latent watermarker achieves competitive robustness across various attacks, with only a slight drop in average bit accuracy (95.2% vs. 97.8%) compared to the pixel watermarker. It maintains strong performance on valuemetric transformations (98.1%), and compression (99.2%), although combined attacks prove more challenging (84.3% vs 97.3%). Inspired by RoSteALS (Bui et al., 2023), we also compare our approach against the coverless method by using our architecture and training, but removing the input dependency. The coverless approach achieves lower robustness (91.8% average accuracy), showing the importance of input-dependent watermarking.

While the PSNR is lower for the latent watermarker (31.1 dB vs. 43.5 dB), this metric is misleading because it measures pixel-level differences rather than perceptual quality. A lower PSNR is expected when operating in latent space, where small perturbations can propagate through the decoder. FID and IS provide more appropriate quality assessments: compared to a baseline FID of 10.66 for non-watermarked images, both pixel and latent watermarkers preserve high visual quality with FID increases of only +0.18 and +0.76, respectively. Inception Score remains similarly high for both methods (99.53 vs 98.11).

The latent watermarker achieves a 20× speedup during in-

ference compared to the pixel watermarker (3m s vs 63 ms per image on CPU) by operating on lower-dimensional representations (8×8×128 vs 512×512×3). This substantial computational advantage makes latent watermarking a compelling alternative, offering competitive robustness and visual quality at significantly reduced inference cost.

**Discrete latent space.** Table 2 shows results for the RAR-XL autoregressive model. We evaluate two variants: watermarking before quantization (applied to continuous latents) and after quantization (applied to token embeddings). Watermarking after quantization achieves higher robustness (93.96% vs. 90.60% average bit accuracy) compared to before quantization. This is because the quantization step maps continuous latents to discrete tokens, removing subtle perturbations; the watermark must be strong enough to alter the token sequence itself. Compared to pixel watermarking, latent watermarking after quantization achieves slightly lower robustness (93.96% vs. 97.27% average bit accuracy).

For visual quality, FID is similar across methods (3.44 for latent vs. 3.09 for pixel), indicating that quality is preserved.

Note that only post-hoc watermarking before the quantization step is suitable for distillation into the autoregressive model. Indeed, the resulting watermarked latents are then quantized into discrete tokens, forming a watermarked sequence of tokens (the tokens themselves are not watermarked) that can be directly used to fine-tune the autoregressive model. On the other hand, post-hoc watermarking after the quantization step modifies the continuous embedding of each token; therefore, it does not produce a new sequence of tokens and cannot be used for distillation into the generative model. However, it can be distilled into the latent decoder.

In terms of efficiency, post-hoc latent watermarking achieves a 2.4× speedup compared pixel watermarking (17m s vs 41 ms per image on CPU) by operating on lower-dimensional representations (16×16×256 vs 256×256×3). The speedup is lower than that of DCAE because the latent space of RAR-XL is less compressed (16×16 vs. 8×8

*Table 3.* Comparing visual quality and robustness after distillation of the post-hoc watermarkers in the latent decoder of DCAE. The first line of each subgroup corresponds to the post-hoc watermarker used as teacher for distillation. The following lines correspond to different extractor weights $\lambda_w$ used during distillation.

| Method | Extractor Weight | PSNR | FID | IS | Avg | Combined |
|---|---|---|---|---|---|---|
| Post Hoc Pixel (PHP) | Teacher | 43.48 | 10.84 | 99.53 | 97.78 | 97.29 |
| Distilled from PHP (1) | 0 | 37.17 | 10.72 | 99.57 | 51.38 | 50.72 |
| Distilled from PHP (2) | 0.1 | 33.65 | 11.37 | 97.45 | 89.16 | 62.16 |
| Distilled from PHP (3) | 1.0 | 30.22 | 17.08 | 80.84 | 92.95 | 65.69 |
| Post Hoc Latent (PHL) | Teacher | 31.06 | 11.42 | 98.11 | 95.18 | 84.28 |
| Distilled from PHL (1) | 0 | 31.39 | 11.34 | 97.75 | 94.41 | 82.58 |
| Distilled from PHL (2) | 0.1 | 31.20 | 11.48 | 97.21 | 96.77 | 91.34 |

spatially, and 256 vs. 128 channels). See more details in Appendix C.

## 5.3. In-model Watermarking Results

### 5.3.1. DISTILLATION IN THE LATENT DECODER

For DCAE and RAR, we distill post-hoc latents and pixel watermarkers into the latent decoder as described in Eq. 3.

**Continuous latent space.** We report in Table 3 the results of distilling both the post-hoc latent and pixel watermarkers into the latent decoder of DCAE when using different extractor weights $\lambda_w$. If we only use the reconstruction loss (i.e., setting the extractor loss weight to 0), we observe that distilling the post-hoc latent watermarker into the latent decoder achieves 94.41% bit accuracy on average over transformations (compared to 95.18% for the teacher model) while preserving the visual quality (with 11.34 FID vs 11.42 for the teacher). This shows that the latent decoder can learn to generate watermarked images when trained to reconstruct the watermarked outputs of the post-hoc latent watermarker. When using the extractor loss, the robustness further improves to 96.77% average bit accuracy, which is even better

than the teacher post-hoc watermarker while maintaining a similar FID compared to the teacher model (11.48 vs 11.42). This demonstrates that incorporating the extractor loss during distillation effectively encourages the latent decoder to produce images with more robust watermarks. On the other hand, the pixel watermarker fails to be distilled in the latent decoder when using the reconstruction loss alone with only 51.38% bit accuracy. When adding a strong weight on the extractor loss, the robustness improves to an average bit accuracy of 92.95%, but remains significantly lower than that of the teacher pixel watermarker (97.78%) or the distilled latent watermarker (96.77%), and it comes at the cost of a much higher FID (17.08). A potential explanation is that the pixel watermarker introduces high-frequency perturbations in the pixel space, which are difficult for the latent decoder to reconstruct due to its limited capacity. In contrast, the latent watermarker introduces more structured perturbations in the latent space, which are easier for the latent decoder to learn. We can see in figure 3 that the distilled latent decoder manages to reproduce the watermark patterns introduced by the post-hoc latent watermarker, whereas the distilled pixel watermarker fails to reproduce the high-frequency watermark patterns of the post-hoc pixel watermarker.

**Discrete latent space.** The post-hoc latent watermarker applied after the quantization step can also be distilled in the latent decoder of the autoregressive model, as shown in Table 4 and illustrated in Figure 4. Without the extractor loss, distillation achieves 81.40% average bit accuracy, which is lower than the teacher post-hoc watermarker (93.96%). When adding the extractor loss during distillation, the robustness further improves to 90.57% average bit accuracy while preserving similar visual quality (3.62 vs 3.44 FID

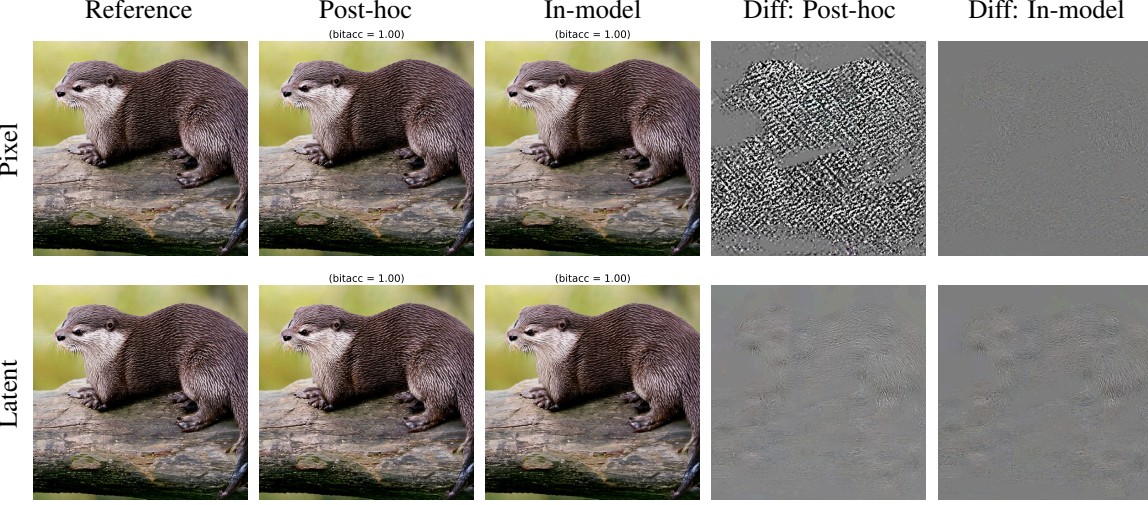

*Figure 3.* We compare pixel (top) and latent (bottom) post-hoc watermarks on a DCAE-generated image (ImageNet class=360). In the second column, we show the watermarked images after applying the post-hoc watermarkers. In the third column, we show in-model outputs after distilling the respective watermarkers into the latent decoder. In the last two columns, we show the difference images between the watermarked and reference images.

*Table 4.* Comparing the visual quality and bit prediction robustness after distillation of the post-hoc watermarking models in the latent decoder of RAR. The first line of each subgroup corresponds to the post-hoc watermarking model used as teacher for distillation. The following lines correspond to different extractor weights used during the distillation process.

| Method | Extractor Weight | PSNR | FID | IS | Avg | Combined |
|---|---|---|---|---|---|---|
| Post Hoc Pixel (PHP) | Teacher | 41.81 | 3.09 | 292.75 | 97.27 | 93.93 |
| Distilled from PHP (1) | 0 | 48.20 | 3.12 | 294.14 | 50.15 | 49.61 |
| Distilled from PHP (2) | 0.1 | 40.64 | 3.15 | 293.65 | 50.81 | 49.69 |
| Distilled from PHP (3) | 1.0 | 22.95 | 5.05 | 263.39 | 71.25 | 49.84 |
| Post Hoc Latent (PHL) | Teacher | 24.54 | 3.44 | 250.50 | 93.96 | 82.35 |
| Distilled from PHL (1) | 0 | 26.77 | 3.57 | 284.23 | 81.40 | 63.01 |
| Distilled from PHL (2) | 0.1 | 26.06 | 3.62 | 281.39 | 90.57 | 69.77 |
| Distilled from PHL (2) | 0.5 | 25.64 | 3.70 | 278.52 | 92.33 | 73.37 |

for the teacher model). Compared to the latent decoder of DCAE, the latent decoder of RAR seems to have a lower capacity, as the distilled latent watermarker cannot reach the same robustness as the teacher post-hoc watermarker even when using the extractor loss during distillation. The post-hoc pixel watermarker fails to be distilled in the latent decoder of RAR with only 50.15% average bit accuracy without the extractor loss and 71.25% with a strong extractor loss weight (and a high FID of 5.05).

In Appendix E, we provide additional comparisons of distilling other post-hoc pixel watermarking methods into the latent decoders of DCAE and RAR, along with ablation studies on our training loss formulation compared to Stable Signature in Appendix F.1.

### 5.3.2. DISTILLATION IN THE GENERATIVE MODEL

Unlike pixel watermarkers, we can use the post-hoc latent watermarkers to change the training latents into watermarked latents and use them to fine-tune the diffusion and autoregressive models while maintaining the watermark-ing robustness. We report in Table 5 the results of distilling the post-hoc latent watermarkers into the generative part of the diffusion model and into the autoregressive model, respectively. By simply fine-tuning on watermarked latents, the distilled generative models achieve comparable robustness compared to their teacher model (94.78% vs 95.18% average bit accuracy for DCAE and 89.13% vs 90.60% for RAR) while having better visual quality metrics than their teacher models (10.90 vs 11.42 FID for DCAE and 3.32 vs 3.56 FID for RAR). It is a remarkable result that the distilled autoregressive model can generate new watermarked sequences of tokens that have the same robustness to transformations as the post-hoc latent watermarker. Similarly, for the diffusion model, the distilled diffusion model can generate new watermarked latents with the same robustness as its teacher. Furthermore, compared to distillation in the latent decoder, distillation in the generative model achieves slightly lower robustness (94.78% vs 96.77% average bit accuracy for DCAE and 89.13% vs 90.57% for RAR) but with better visual quality (10.90 vs 11.48 FID for DCAE and 3.32 vs 3.62 FID for RAR). This suggests that both distillation approaches are effective for in-model watermarking.

### 5.4. Analysis of Diffusion DISTSEAL

We analyze how the watermark is formed during the diffusion process when using the distilled diffusion model. As illustrated in Figure 5, we observe that the bit accuracy progressively increases throughout the diffusion process, demonstrating that the watermark strengthens as the image is refined. Figure 16 illustrates this progressive watermark formation across different diffusion steps.

In the right plot of Figure 5, we examine a hybrid approach

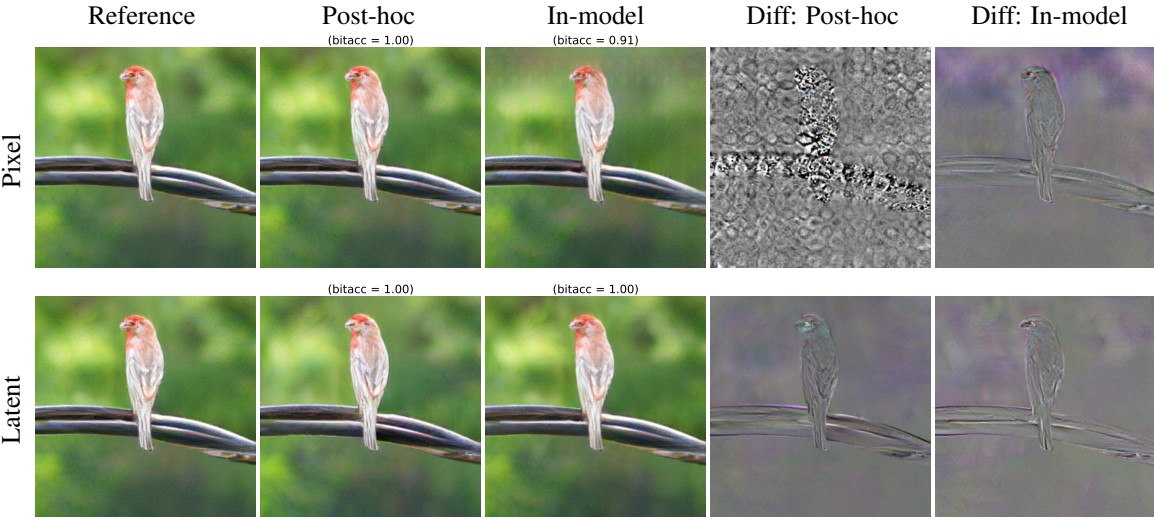

*Figure 4.* We compare pixel (top) and latent (bottom) post-hoc watermarks on a RAR-generated image (ImageNet class=975). In the second column, we show the watermarked images after applying the post-hoc watermarkers. In the third column, we show in-model outputs after distilling the respective watermarkers into the latent decoder. Here, the post-hoc latent watermarker is applied after the quantization step. In the last two columns, we show the difference images between the watermarked and reference images.

*Table 5.* Distillation results for DCAE and RAR. The lines in gray correspond to the post-hoc latent watermarkers used as teacher for distillation. The following lines correspond to either distilling into the generative model or the latent decoder. For RAR, the post-hoc latent watermarker before quantization is used for distillation into the generative model, while the post-hoc latent watermarker after quantization is used for distillation into the latent decoder.

| DCAE | FID | IS | Identity | Valuemetric | Geometric | Compression | Combined | Avg |
|---|---|---|---|---|---|---|---|---|
| Post hoc Latent | 11.42 | 98.11 | 99.75 | 98.08 | 91.62 | 99.23 | 84.28 | 95.18 |
| In-model (diffusion model) | 10.90 | 101.25 | 99.53 | 97.67 | 91.27 | 98.88 | 83.37 | 94.78 |
| In-model (latent decoder) | 11.48 | 97.21 | 99.99 | 99.36 | 93.35 | 99.73 | 91.34 | 96.77 |

| RAR | FID | IS | Identity | Valuemetric | Geometric | Compression | Combined | Avg |
|---|---|---|---|---|---|---|---|---|
| Post hoc (before quantization) | 3.56 | 239.00 | 96.92 | 95.91 | 87.20 | 95.98 | 77.00 | 90.60 |
| In-model (autoregressive model) | 3.32 | 288.22 | 94.85 | 93.88 | 85.70 | 93.92 | 77.30 | 89.13 |
| Post hoc (after quantization) | 3.43 | 250.50 | 99.52 | 98.56 | 91.37 | 97.97 | 82.35 | 93.96 |
| In-model (latent decoder) | 3.62 | 281.39 | 99.09 | 97.65 | 89.95 | 96.37 | 69.77 | 90.57 |

where the original diffusion model handles the initial denoising steps before switching to the distilled model for the final $N$ steps. Remarkably, even when the distilled model is applied only during the last few diffusion steps, we maintain high bit accuracy, suggesting that watermark information can be effectively embedded in the final refinement stage.

Unlike post-hoc watermarking, where we can directly visualize watermark patterns through pixel differences, the distilled model generates watermarked images directly from random noise, making direct visualization of the watermark patterns challenging. However, when employing the distilled model for only the final 5 steps, the resulting images remain visually similar to those from the reference model, enabling watermark visualization through difference images. Figure 18 presents several examples of these visualized watermark patterns. In Appendix K, we provide some analysis of autoregressive DISTSEAL.

### 5.5. How to Choose Between Distilling in the Generative Model or in the Latent Decoder?

Both distillation strategies offer distinct advantages and limitations. Table 6 summarizes the trade-offs when distilling the watermarker in the generative model versus the latent decoder. The choice between these approaches depends on the specific deployment scenario. Distilling in the generative model is preferable when a non-watermarked version of the latent decoder is already publicly available or when deployment simplicity and visual quality are prioritized. Distilling

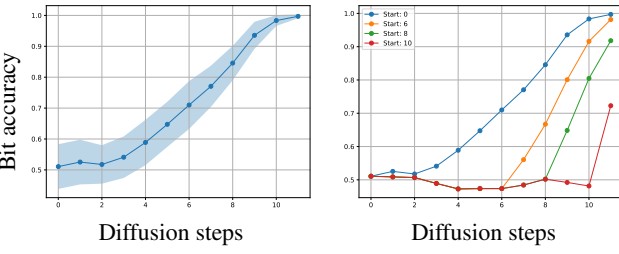

*Figure 5.* Bit accuracy over diffusion steps: detection improves during the generation process, using the distilled model from the beginning (left), or only for the last N diffusion steps (right).

*Table 6.* Generative model vs. latent decoder distillation.

| Criterion | Generative Model | Latent Decoder |
|---|---|---|
| Ease of optimization | Plug and play. No hyperparameters to tune, simply fine-tuning on watermarked latents. | Needs to tune extractor weight $\lambda_w$ and LPIPS weight $\lambda_p$ to trade-off quality and detection. |
| Visual quality | Same or better than the teacher model | Higher FID and risk of blurry decoding if $\lambda_p$ not properly tuned |
| Watermark detection | Retains the teacher's robustness | Retains the teacher's robustness |
| Watermark forgetting | Susceptible to forgetting with LoRA. | Not affected by changes on the generative model |
| Latent decoder switching | Robust to the latent decoder fine-tuning | Vulnerable to decoder replacement |
| Computation overhead | None during inference | None during inference |
| Flexibility | Fixed WM message | Fixed WM message |

in the latent decoder is advantageous when the generative model is frequently updated or when stronger persistence against fine-tuning attacks is required.

## 6. Conclusion and Limitations

We introduced DISTSEAL, a framework for watermarking in the latent space of diffusion and autoregressive generative models. We show that post-hoc latent watermarking can achieve competitive robustness compared to pixel watermarking while being significantly faster for compressed latent spaces, such as DCAE. Furthermore, we demonstrate that post-hoc latent watermarkers can be effectively distilled into the latent decoder or the generative model itself, enabling in-model watermarking and facilitating open-sourced deployment of watermarked generative models.

However, as with any watermarking technology, it is still possible to remove watermarks, as shown in Section 6.3. Furthermore, the distillation method is bounded by the post-hoc watermarker performance (as shown in the appendix), and in the case of autoregressive model, it is further bounded by the number of discrete tokens. Our method is still not robust enough against very strong attacks such as some geometric transformations, for which we are exploring the post-hoc synchronisation methods (Fernandez et al., 2025).

As future work, we plan to extend our method to video generation, as latent watermarking could offer a significant speedup compared to pixel watermarking every frame.

## Impact Statement

The ability to watermark generated images is crucial for establishing content provenance and protecting intellectual property in the era of AI-generated media. DISTSEAL provides a practical and efficient solution for embedding robust watermarks directly within generative models, facilitating open-sourced deployment while minimizing watermarking overhead.

As a practical guideline for selecting the most suitable approach, latent post-hoc watermarking is preferable when the watermark will be distilled into a model or for highly efficient applications like video. Conversely, pixel post-hoc watermarking is ideal when maximum robustness under extreme combined attacks is required, or when watermarking non generated content. Finally, in-model watermarking via DistSeal distillation is best when: (a) the model is open-source and watermark removal must be prevented, (b) zero inference overhead is required, or (c) deployment simplicity is prioritized.

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

# A. Implementation Details

**Post-hoc watermarking training.** We train all posthoc watermarkers for 600k steps using the Adam optimizer with a ramp-up of 20k steps to a learning rate of 5e-4 followed by a cosine decay. We use a batch size of 128. In Equation 1, we set the weights $\lambda_w = 1.0$ and $\lambda_{\text{disc}} = 0.1$. We use the discriminator from (Weber et al., 2024) with 3 layers and we only start using the discriminator loss after 200k steps. For all the models, the watermark strength is set to higher value for the first 100k steps to kickstart training and then it is decreased to its final value over the next 100k steps with a cosine decay. For the latent watermarker of DCAE, the watermark strength goes from $\epsilon = 1.5$ to $\epsilon = 0.5$. For the latent watermarker of RAR, it starts at $\epsilon = 2.0$ to decrease at $\epsilon = 0.5$ when the latent watermarker is after the quantization step. If the latent watermarker is before the quantization step, it starts at $\epsilon = 3.0$ to decrease at $\epsilon = 1.5$. Finally, for the pixel watermarker, it goes from $\epsilon = 0.2$ to $\epsilon = 0.02$.

**Autoregressive generator distillation training.** We use the same training setting used to finetune RAR-XL (Yu et al., 2025), (effective batch size 2048 distributed over 32 GPUs, data are augmented with random flip and tencrop and are resized to 256x256 resolution). The only difference from the original RAR-XL training is that we set the learning rate to 1e-5 with zero warmup and a constant scheduler instead of cosine. This mimics the scenario where one continues the training of original RAR-XL after reaching stationary learning rate, but with watermarked tokens as targets. The model was trained for 10k steps (about 2 epochs over the augmented dataset), which took 50 GPU-hours. We experiment with different watermark strengths $\epsilon$ (0.7, 1.0 and 1.5).

# B. Evaluation Details

For the quality metrics, we report FID score using torch-fidelity implementation [1] , computed over 50k generated samples against the ImageNet-1k validation set. For the posthoc watermarking methods, we also compute the PSNR and for the watermark detection metrics we report the bit accuracy over the transformations described below.

**Transformations.** The watermarked images are evaluated under a series of transformations, shown in Table 7. For each parameter value, we apply the transformation to the entire set of 50,000 generated images. The reported *average* bit accuracy is obtained by averaging the scores computed for each transformation category.

*Table 7.* Transformations applied during evaluation.

| Transformation | Category | Parameters |
|---|---|---|
| Identity | Identity | |
| Brightness Adjustment | ValueMetric | factor: [0.1, 0.25, 0.5, 0.75, 1.0, 1.25, 1.5, 1.75, 2.0] |
| Horizontal Flip | Geometric | |
| Rotate | Geometric | angle: [5, 10, 30, 45, 90] |
| Resize | Geometric | size: [0.32, 0.45, 0.55, 0.63, 0.71, 0.77, 0.84, 0.89, 0.95, 1.00] |
| Crop | Geometric | size: [0.32, 0.45, 0.55, 0.63, 0.71, 0.77, 0.84, 0.89, 0.95, 1.00] |
| Contrast | ValueMetric | factor: [0.1, 0.25, 0.5, 0.75, 1.0, 1.25, 1.5, 1.75, 2.0] |
| Saturation | ValueMetric | factor: [0.1, 0.25, 0.5, 0.75, 1.0, 1.25, 1.5, 1.75, 2.0] |
| Gray scale | ValueMetric | |
| Hue | ValueMetric | factor: [-0.4, -0.3, -0.2, -0.1, 0.0, 0.1, 0.2, 0.3, 0.4, 0.5] |
| JPEG Compression | Compression | quality: [40, 50, 60, 70, 80, 90] |
| Gaussian Blur | ValueMetric | kernel size: [3, 5, 9, 13, 17] |
| JPEG - Crop - Brightness | Combined | (jpeg quality=[40, 60, 80], crop size=0.71, brightness factor=0.5) |

---

[1]https://github.com/toshas/torch-fidelity

## C. Wall-clock Time Comparison

We provide more details on the wall-clock time benchmark of the watermarking within the full generation pipeline. For each generative model (DCAE and RAR-XL), we generate 100 images on a single NVIDIA H200 GPU. The result is shown in Table 8. We report the time for each stage of the generation pipeline, including the time for post-hoc watermarking and the total time. We also compute the watermarking overhead as a percentage of the total time.

*Table 8.* Wall-clock Time Benchmarking of DC-AE and RAR on a single H200 GPU

| Stage | Time (ms) | With pixel WM | With latent WM |
|---|---|---|---|
| DCAE (512×512 images) | | | |
| Diffusion generation (250 steps) | ∼442 | | |
| Latent decoding | ∼15 | n.a. | ∼15 |
| Post-hoc watermarking | | ∼5 | ∼2 |
| Total | ∼6457 | ∼6462 | ∼6459 |
| Watermarking overhead | | 0.1% | < 0.1% |
| RAR-XL (256×256 images) | | | |
| Token generation (256 tokens) | | ∼2623 | |
| Latent decoding | ∼5 | n.a. | ∼5 |
| Post-hoc watermarking | | ∼3 | ∼1 |
| Total | ∼2628 | ∼2631 | ∼2629 |
| Watermarking overhead | | 0.1% | < 0.1% |

We observe that for both diffusion (DC-AE) and autoregressive (RAR-XL) pipelines, watermarking adds negligible overhead ($< 0.2\%$) since generation dominates runtime. Note that for in-model watermarking (distilled into the generative model or decoder), the watermarking overhead is exactly 0 ms as the watermark is embedded as part of the normal generation process with no additional computation so this would avoid the latency cost of calling a watermarking API. Our method would be more promising for example the video setting, as watermarking every frame would accumulate a non negligible latency overhead on top of generation.

## D. Impact of Teacher-guided Distillation

In theory, one can also traing the direct in-model watermarking, i.e. to combine two steps in one training and train the latent encoder or decoder to directly reconstruct the watermarked image. However, we argue that using teacher-guided distillation offers the following benefits:

- *Decoupled optimization.* Training the watermarker separately lets us tune robustness and imperceptibility without complicating the generative objective. Joint training would require balancing watermark losses against generation losses throughout training, making optimization harder.

- *Reusability and flexibility.* One post-hoc watermarker can act as a teacher for multiple generative models that share the same autoencoder/latent space (e.g., a DCAE latent watermarker distilled into UViT-H, DiT, etc.). Direct training would need separate end-to-end training per model.

For empirical validation, we train watermarking from scratch by jointly fine-tuning the DCAE diffusion model and a watermark extractor, and compare with our distillation approach. The resulting bit-accuracy results are reported in Table 9. We can see that the direct in-model training approach performs significantly worse than the teacher-guided distillation approach, especially on more complex transformations such as geometric and combined transformations.

*Table 9.* Comparison between teacher-guided distillation and direct in-model training.

| Method | FID | Identity | Value Metric | Geometric | Compression | Combined | Avg |
|---|---|---|---|---|---|---|---|
| DistSeal (teacher-guided distillation) | 11.48 | 99.99 | 99.36 | 93.35 | 99.73 | 91.34 | 96.77 |
| Direct in-model training (no teacher) | 31.35 | 99.12 | 96.33 | 91.34 | 97.62 | 78.34 | 92.55 |

# E. Distillation in the latent decoder

We provide additional comparisons of distilling various post-hoc watermarking methods into the latent decoder of DCAE and RAR-XL in Table 10 and Table 11 respectively. We observe that it is much tougher to distill into the latent decoder of RAR than DCAE. In fact, besides the post-hoc latent watermarker, only WAM can be distilled in the latent decoder of RAR. For DCAE, all the methods except for the post-hoc pixel watermarker and MBRS can even be distilled with just the reconstruction loss (i.e. without any extractor loss ($\lambda_w = 0$)).

*Table 10.* Distilling various post-hoc watermarking methods in the latent decoder of DCAE. For each watermarking method, we report the results of the teacher post-hoc watermarker and the distilled model in terms of visual quality (FID) and bit accuracy over transformations.

| Method | Distillation | FID | Identity | Valuemetric | Geometric | Compression | Combined | Avg |
|---|---|---|---|---|---|---|---|---|
| Post-hoc pixel | Teacher | 10.84 | 100.00 | 99.89 | 94.70 | 99.90 | 97.29 | 97.78 |
| | $\lambda_w = 0$ | 10.72 | 51.58 | 51.56 | 51.14 | 51.78 | 50.72 | 51.38 |
| | $\lambda_w = 0.1$ | 11.37 | 99.99 | 93.65 | 87.35 | 78.92 | 62.16 | 89.16 |
| Post-hoc latent | Teacher | 11.42 | 99.75 | 98.08 | 91.62 | 99.23 | 84.28 | 95.18 |
| | $\lambda_w = 0$ | 11.34 | 99.19 | 97.33 | 90.91 | 98.39 | 82.58 | 94.41 |
| | $\lambda_w = 0.1$ | 11.48 | 99.99 | 99.36 | 93.35 | 99.73 | 91.34 | 96.77 |
| CIN | Teacher | 11.09 | 99.99 | 88.55 | 64.86 | 99.27 | 48.68 | 78.78 |
| | $\lambda_w = 0$ | 10.89 | 98.81 | 86.72 | 64.74 | 94.28 | 48.61 | 77.48 |
| | $\lambda_w = 0.1$ | 11.09 | 99.94 | 87.64 | 65.11 | 99.22 | 48.70 | 78.43 |
| MBRS | Teacher | 10.99 | 97.49 | 91.72 | 64.66 | 96.27 | 49.66 | 80.08 |
| | $\lambda_w = 0$ | 10.87 | 51.22 | 50.89 | 50.07 | 51.14 | 49.56 | 50.54 |
| | $\lambda_w = 0.1$ | 10.96 | 99.99 | 95.27 | 65.81 | 99.85 | 49.66 | 82.55 |
| TrustMark | Teacher | 11.28 | 99.78 | 97.15 | 75.61 | 98.94 | 51.68 | 87.31 |
| | $\lambda_w = 0$ | 11.11 | 94.57 | 90.66 | 72.17 | 92.05 | 50.82 | 82.20 |
| | $\lambda_w = 0.1$ | 11.51 | 99.99 | 97.72 | 75.89 | 99.33 | 51.42 | 87.73 |
| WAM | Teacher | 11.61 | 100.00 | 91.01 | 86.51 | 99.07 | 54.97 | 88.67 |
| | $\lambda_w = 0$ | 11.39 | 92.98 | 84.52 | 79.03 | 92.30 | 50.33 | 81.83 |
| | $\lambda_w = 0.1$ | 11.32 | 100.00 | 89.80 | 82.73 | 97.11 | 49.01 | 86.25 |

*Table 11.* Distilling various post-hoc watermarking methods in the latent decoder of RAR. For each watermarking method, we report the results of the teacher post-hoc watermarker and the distilled model in terms of visual quality (FID) and bit accuracy over transformations.

| Method | Distillation | FID | Identity | Valuemetric | Geometric | Compression | Combined | Avg |
|---|---|---|---|---|---|---|---|---|
| Post-hoc pixel | Teacher | 3.09 | 100 | 99.69 | 92.97 | 99.74 | 93.93 | 97.27 |
| | $\lambda_w = 0$ | 3.12 | 50.15 | 50.12 | 50.32 | 50.57 | 49.61 | 50.15 |
| | $\lambda_w = 0.1$ | 3.15 | 51.12 | 50.89 | 50.80 | 51.53 | 49.69 | 50.81 |
| Post-hoc latent (before quantization) | Teacher | 3.56 | 96.92 | 95.91 | 87.20 | 95.98 | 77.00 | 90.60 |
| | $\lambda_w = 0$ | 3.99 | 57.06 | 56.53 | 55.06 | 56.26 | 50.71 | 55.12 |
| | $\lambda_w = 0.1$ | 3.99 | 51.94 | 53.10 | 51.55 | 55.24 | 50.73 | 52.51 |
| Post-hoc latent (after quantization) | Teacher | 3.44 | 99.52 | 98.56 | 91.37 | 97.97 | 82.35 | 93.96 |
| | $\lambda_w = 0$ | 3.57 | 89.92 | 87.45 | 80.58 | 86.03 | 63.01 | 81.40 |
| | $\lambda_w = 0.1$ | 3.62 | 99.09 | 97.65 | 89.95 | 96.37 | 69.77 | 90.57 |
| CIN | Teacher | 3.11 | 100 | 88.87 | 58.22 | 92.32 | 47.32 | 77.35 |
| | $\lambda_w = 0$ | 3.11 | 49.12 | 48.65 | 49.05 | 48.18 | 48.45 | 48.69 |
| | $\lambda_w = 0.1$ | 3.14 | 56.79 | 53.56 | 50.50 | 50.02 | 48.41 | 51.86 |
| MBRS | Teacher | 3.09 | 100 | 94.29 | 59.51 | 99.63 | 49.58 | 80.60 |
| | $\lambda_w = 0$ | 3.10 | 50.08 | 49.70 | 49.68 | 50.18 | 49.25 | 49.78 |
| | $\lambda_w = 0.1$ | 3.10 | 51.39 | 50.69 | 49.90 | 50.96 | 49.23 | 50.43 |
| TrustMark | Teacher | 3.28 | 99.91 | 97.67 | 77.47 | 98.18 | 52.18 | 85.08 |
| | $\lambda_w = 0$ | 3.20 | 49.96 | 50.04 | 50.08 | 50.12 | 50.27 | 50.09 |
| | $\lambda_w = 0.1$ | 3.23 | 50.94 | 50.86 | 50.63 | 50.67 | 50.53 | 50.73 |
| WAM | Teacher | 3.42 | 100 | 86.85 | 86.49 | 95.25 | 44.75 | 82.67 |
| | $\lambda_w = 0$ | 3.09 | 64.24 | 60.28 | 56.05 | 60.94 | 47.55 | 57.81 |
| | $\lambda_w = 0.1$ | 3.16 | 99.47 | 84.01 | 67.24 | 76.39 | 45.40 | 74.50 |

# F. Comparison with In-model Watermarking Methods

### F.1. Stable Signature

In this Section, we compare our distillation approach with other in-model watermarking methods, focusing on diffusion. The closest one to ours is Stable Signature, which trains a post-hoc watermarking model, but during distillation, it discards the teacher's embedder and uses only the watermark extractor as a guidance signal for the distillation into the latent decoder. In contrast, our method retains the teacher embedder during distillation and trains the latent decoder to reconstruct the watermarked image. We do a direct comparison between the two methods by using Stable Signature's loss (i.e. replacing $\mathcal{L}_{rec}\left(\mathcal{D}_o(z_w), \mathcal{D}(z)\right)$ by $\mathcal{L}_{rec}\left(\mathcal{D}_o(z), \mathcal{D}(z)\right)$ in Eq. 3) to distill our post-hoc latent watermarking model into the latent decoder of DCAE.

The results are shown in Table 12, where we observe that Stable Signature performs similarly as the teacher model on valuemetric and geometric transformations. However, it performs significantly worse on combined transformations with 77.65% bit accuracy even when increasing the extractor weight to $\lambda_w = 0.5$ compared to 84.28% for the teacher model. In contrast, DISTSEAL reaches 82.58% bit accuracy on combined transformations even with $\lambda_w = 0$ (i.e., without extractor guidance), and further improves to 91.34% with $\lambda_w = 0.1$. This shows that the embedder reconstruction term of DISTSEAL allows to preserve the robustness of the teacher model on all covered transforms.

Furthermore, we notice that increasing the extractor weight beyond a certain point (like $\lambda_w = 0.5$ for DISTSEAL or $\lambda_w = 1.0$ for Stable Signature) continues to improve watermark detection on unmodified images (Identity) but degrades robustness against all the transformations for both Stable Signature and our method. This suggests that overemphasizing the extractor loss may lead to overfitting to unmodified images at the cost of robustness. Visual comparisons are shown in Figure 6. We see that the distilled watermark with Stable Signature does not look similar to the original watermark, while our method preserves the watermark patterns of the teacher model.

*Table 12.* Comparing Stable Signature with DISTSEAL for distilling the same post-hoc latent watermarking model (in gray) into the latent decoder of DCAE. We sweep over several extractor weight $\lambda_w$ and report the bit accuracy under different transformations as well as the FID. We observe that Stable Signature does not match the robustness of DISTSEAL on more complex transformations such as "Combined".

| Method | Distillation | FID | Identity | Value Metric | Geometric | Compression | Combined | Avg |
|---|---|---|---|---|---|---|---|---|
| Post-hoc latent | - | 11.42 | 99.75 | 98.08 | 91.62 | 99.23 | 84.28 | 95.18 |
| DISTSEAL | $\lambda_w = 0$ | 11.34 | 99.19 | 97.33 | 90.91 | 98.39 | 82.58 | 94.41 |
| | $\lambda_w = 0.01$ | 11.36 | 99.88 | 98.77 | 92.65 | 99.40 | 88.07 | 96.07 |
| | $\lambda_w = 0.1$ | 11.48 | 99.99 | 99.36 | 93.35 | 99.73 | 91.34 | 96.77 |
| | $\lambda_w = 0.5$ | 13.42 | 100.00 | 91.68 | 81.26 | 82.97 | 69.81 | 86.35 |
| Stable Signature | $\lambda_w = 0.1$ | 11.39 | 99.96 | 98.51 | 90.83 | 97.60 | 73.24 | 94.59 |
| | $\lambda_w = 0.5$ | 12.09 | 100.00 | 99.16 | 91.61 | 98.61 | 77.65 | 95.44 |
| | $\lambda_w = 1.0$ | 14.14 | 100.00 | 88.28 | 73.63 | 71.93 | 58.58 | 80.54 |

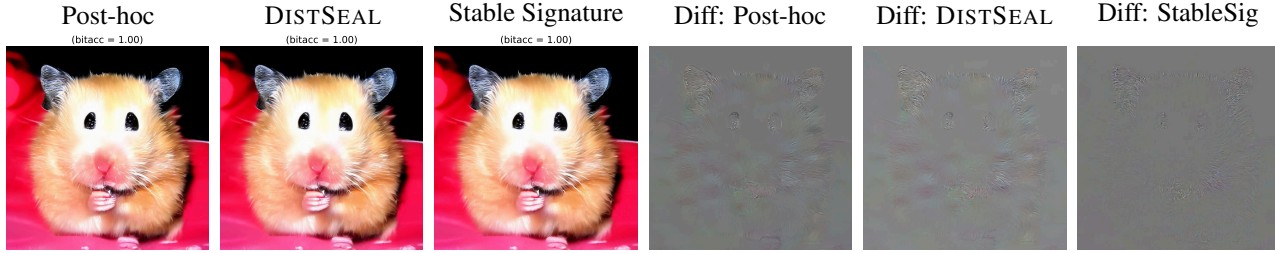

*Figure 6.* We compare the watermarks when distilling the same post-hoc latent watermarking model into the latent decoder of DCAE using DISTSEAL and Stable Signature respectively. The first column shows the watermarked images generated by the post-hoc model, followed by watermarked images generated by DISTSEAL (second column) and Stable Signature (third column). The last three columns show the respective diff images. We can see that DISTSEAL preserves the watermark pattern of the teacher model much better than Stable Signature.

### F.2. Other In-model Watermarking Models

We also compare DISTSEAL with other non-distillation methods such as RoSteALS (Bui et al., 2023) and Tree-Ring (Wen et al., 2023). RoSteALS is a steganography method using autoencoder latent space, while Tree-Ring is a method that an in-model method that inject a carefully structured pattern in the initial noise vector's Fourier space of the diffusion . Here we report both TPR@FPR=0.01 and bit accuracy for RoSTeALS, while for Tree-Ring, it is a zero-bit method, so we report only the TPR@FPR=0.01. For reference, we compar against DISTSEAL with $\lambda_w = 0.1$ as it is the best performing model in the distillation setting. The results are shown in Table 13. We observe that DISTSEAL significantly outperforms both RoSteALS and Tree-Ring in terms of robustness against transformations, especially for geometric and combined transformations, while achieving comparable performance on unmodified images (Identity).

*Table 13.* Comparison of DCAE-distilled latent decoder with RoSteALS and Tree-Ring (TPR@FPR=0.01 — Bit Accuracy).

| Method | Identity | Valuemetric | Geometric | Compression | Combined | Avg |
|---|---|---|---|---|---|---|
| DISTSEAL ($\lambda_w = 0.1$) | 1.00 — 99.99 | 0.99 — 99.36 | 0.92 — 93.35 | 1.00 — 99.73 | 0.96 — 91.34 | 0.97 — 96.77 |
| RoSteALS (coverless) | 1.00 — 99.96 | 0.55 — 98.51 | 0.41 — 90.83 | 1.00 — 97.60 | 0.01 — 73.24 | 0.49 — 94.59 |
| Tree-Ring | 0.99 | 0.82 | 0.71 | 0.99 | 0.27 | 0.76 |

## G. Watermark Forgetting

We investigate the susceptibility of distilled diffusion and autoregressive models to forgetting the watermark when fine-tuned on non-watermarked data. A practical threat scenario for generative models would be a user who wants to perform LoRA fine-tuning (Hu et al., 2022) on their own data to tailor the model to their specific needs. As the generative models in our study are class-conditional, we simulate the LoRA scenario by fine-tuning both distilled diffusion and autoregressive models

on a single class using a small set of 50 non-watermarked ImageNet images from the validation set.

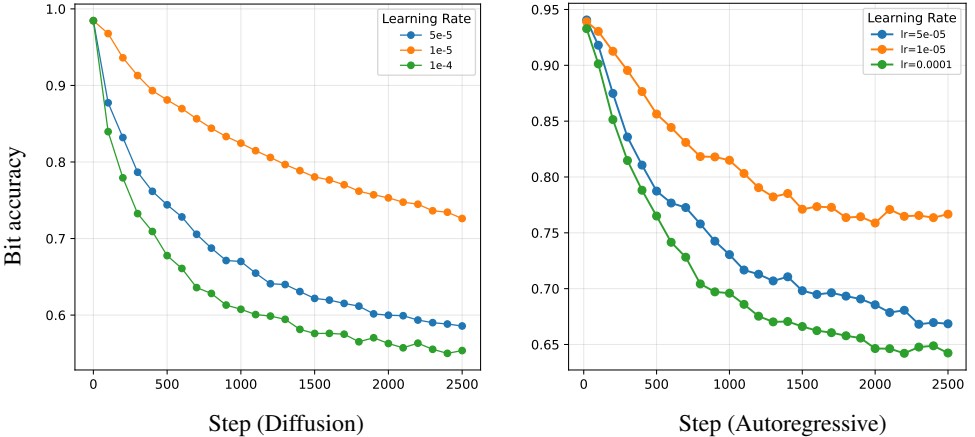

*Figure 7.* Mean bit accuracy over LoRA-finetuning steps, trained on 50 non-watermarked images of a single ImageNet class. The left and right plots correspond to LoRA-finetuning of distilled diffusion (DCAE) and autoregressive (RAR-XL) models, respectively. Different colors correspond to different learning rates.

Figure 7 shows the bit accuracy over LoRA fine-tuning steps for both distilled diffusion (DC-AE) and autoregressive (RAR-XL) models. For small learning rates like 1e-5 (in orange), the model still retains some watermark detection performance after 2500 steps with 0.82% and 0.76% bit accuracy for the distilled diffusion and autoregressive models, respectively. However, with more aggressive fine-tuning (lr=1e-4), the bit accuracies drop to 0.70% (DC-AE) and 0.65% (RAR-XL). This indicates that distilling into the generative transformers is susceptible to forgetting the watermark when fine-tuned on non-watermarked data, which is a limitation compared to distillation into the latent decoder, as the latter remains unaffected by changes to the generative model.

## H. Multi-watermarking

We analyze whether our in-model watermark can coexist with other watermarking schemes, such as generation-time and post-hoc watermarking. For post-hoc watermarking, we apply various watermarking methods, such as CIN, MBRS, TrustMark, or WAM, to the images generated by the distilled generative models. Generation-time watermarking modifies the sampling procedure of the distilled generative model to incorporate the watermark. For this type of watermark, we only consider the distilled autoregressive model and we choose WMAR (Jovanović et al., 2025) as a generation-time watermarking method to sample the tokens from the distilled autoregressive model. For WMAR, the main hyperparameters are the context window size $h$, the watermark strength $\delta$ and the proportion of the green tokens $\gamma$. We use $h = 1$, $\delta = [0.5, 2.0, 4.0]$ and $\gamma = 0.25$ in the experiments. Tables 14 and 15 show the results of combining the distilled models with various post-hoc watermarking methods (and generation-time method for RAR-XL). We observe that the distilled RAR-XL can coexist with other watermarking methods, resulting in a decrease in average bit accuracy of less than 1 point. However, we note that the distilled DCAE suffers from a greater degradation in watermark detection, with a decrease between 7 and 8 points in average bit accuracy for all the post-hoc watermarking methods.

*Table 14.* Multi-watermarking evaluation results for the distilled RAR-XL models (distilled transformer and distilled latent decoder). The first row of each section shows the results of only using the distilled RAR-XL model without any other watermarker. Each subsequent row shows the change in % bit accuracy after combining with another watermarking method (compared to the first row). For the distilled RAR-XL transformer, we present results with both post-hoc and generation-time watermarking methods. For the distilled latent decoder, only post-hoc watermarking methods are applicable.

| In-model | Auxiliary | Identity | Valuemetric | Geometric | Compression | Combined | Avg |
|---|---|---|---|---|---|---|---|
| | (none) | 94.85 | 93.88 | 85.70 | 93.92 | 77.30 | 89.13 |
| | CIN | -0.51 | -0.61 | -0.57 | -0.32 | -0.34 | -0.47 |
| | MBRS | -0.42 | -0.46 | -0.46 | -0.28 | -0.50 | -0.42 |
| Distilled Transformer | TrustMark | -0.62 | -0.71 | -0.71 | -0.44 | -1.28 | -0.75 |
| | WAM | -0.31 | -0.65 | -0.39 | -0.09 | -0.61 | -0.41 |
| | WMAR ($\delta = 0.5$) | -0.01 | -0.02 | -0.04 | 0.00 | -0.17 | -0.05 |
| | WMAR ($\delta = 2.0$) | -0.22 | -0.22 | -0.20 | -0.17 | -0.38 | -0.24 |
| | WMAR ($\delta = 4.0$) | -0.64 | -0.64 | -0.57 | -0.53 | -0.97 | -0.67 |
| | (none) | 99.09 | 97.65 | 89.95 | 96.37 | 69.77 | 90.57 |
| | CIN | -1.48 | -2.30 | -3.41 | -1.05 | -0.40 | -1.73 |
| Distilled Latent Decoder | MBRS | -0.94 | -1.36 | -1.65 | -0.90 | -0.73 | -1.11 |
| | TrustMark | -0.58 | -0.94 | -1.03 | -0.84 | -1.11 | -0.90 |
| | WAM | -0.54 | -1.47 | -1.51 | -0.26 | -0.53 | -0.86 |

*Table 15.* Multi-watermarking evaluation results for the distilled DCAE models (distilled diffusion and distilled latent decoder). The first row of each section shows the results of only using the distilled DCAE model without any other watermarker. Each subsequent row shows the change in % bit accuracy after combining with another watermarking method (compared to the first row).

| In-model | Auxiliary | Identity | Valuemetric | Geometric | Compression | Combined | Avg |
|---|---|---|---|---|---|---|---|
| | (none) | 99.53 | 97.67 | 91.27 | 98.88 | 83.37 | 94.78 |
| Distilled Diffusion | CIN | -0.54 | -2.97 | -4.27 | -7.40 | -20.63 | -7.16 |
| | MBRS | -1.19 | -4.27 | -4.77 | -8.90 | -20.61 | -7.95 |
| | TrustMark | -0.98 | -3.50 | -4.61 | -8.36 | -20.39 | -7.57 |
| | WAM | -0.62 | -3.71 | -4.69 | -7.63 | -21.46 | -7.62 |
| | (none) | 99.99 | 99.50 | 94.16 | 99.72 | 92.29 | 97.13 |
| Distilled Latent Decoder | CIN | -0.12 | -1.57 | -3.20 | -2.32 | -22.67 | -6.69 |
| | MBRS | -0.52 | -2.61 | -3.55 | -3.20 | -23.04 | -7.30 |
| | TrustMark | -0.19 | -1.75 | -3.56 | -3.16 | -22.78 | -7.00 |
| | WAM | -0.16 | -2.42 | -3.49 | -2.34 | -23.57 | -7.11 |

# I. Additional Experiments with MaskBit

Maskbit (Weber et al., 2024) builds on the improved VQGAN+ tokenizer by adding Lookup-Free Quantization (LFQ). Like for RAR, we would like to learn post-hoc latent watermarkers for this new type of latent (embedding-free bit representation) and then distill the post-hoc latent watermarkers into the autoregressive transformer and the latent decoder. However, we notice that for this type of latent representation, it is not easy to learn a post-hoc watermarker at the bottleneck just before the quantization step. In fact, we can see in the results shown in Table 16, that the post-hoc latent watermarker learned at the bottleneck before quantization (row 2) only reaches 78.06% average bit accuracy. So instead, we learn a post-hoc latent watermarker before the bottleneck projection from 512 channels to 14 channels (row 1), which achieves 88.72% average bit accuracy. We then distill this post-hoc latent watermarker into the autoregressive transformer of MaskBit, reaching 80.56% average bit accuracy (row 3). Next, we learn post-hoc latent watermarkers after the quantization step, either at the bottleneck (row 4) or after the projection from 14 channels to 512 channels (row 5). These two post-hoc latent watermarkers achieve higher average bit accuracy of 95.50% and 92.83% respectively. We then distill the post-hoc latent watermarker learned at the bottleneck after quantization (row 4) into the latent decoder of MaskBit, reaching 90.57% average bit accuracy (row 6). We provide visual comparisons between the post-hoc latent watermarker after quantization (row 4) and its distilled latent decoder (row 6) in Figure 8.

*Table 16.* Distillation results for MaskBit. The first two rows correspond to post-hoc latent watermarkers learned before the quantization step, respectively before the bottleneck projection from 512 channels to 14 channels and at the bottleneck. Similarly, rows 4 and 5 correspond to post-hoc latent watermarkers learned after the quantization step, respectively at the bottleneck and after projection from 14 channels to 512 channels. The lines in gray correspond to the post-hoc latent watermarkers used as teacher for distillation either in the autoregressive transformer for the first group of rows or the latent decoder for the second group of rows.

| Method | Before/after quantization | FID | IS | Identity | Valuemetric | Geometric | Compression | Combined | Avg |
|---|---|---|---|---|---|---|---|---|---|
| Post hoc Latent (before bottleneck) | Before | 3.79 | 258.64 | 95.64 | 94.63 | 86.76 | 94.48 | 72.10 | 88.72 |
| Post hoc Latent (at bottleneck) | Before | 4.36 | 224.90 | 85.42 | 84.38 | 77.27 | 84.28 | 58.97 | 78.06 |
| In-model (autoregressive transformer) | - | 3.57 | 310.32 | 86.40 | 85.88 | 79.49 | 85.79 | 65.25 | 80.56 |
| Post hoc Latent (at bottleneck) | After | 3.09 | 285.14 | 99.78 | 99.78 | 92.03 | 99.50 | 86.90 | 95.50 |
| Post hoc Latent (after bottleneck) | After | 3.05 | 293.34 | 99.25 | 98.52 | 89.63 | 97.87 | 78.90 | 92.83 |
| In-model (latent decoder) | - | 3.43 | 305.52 | 96.95 | 95.98 | 83.52 | 95.49 | 57.64 | 90.57 |

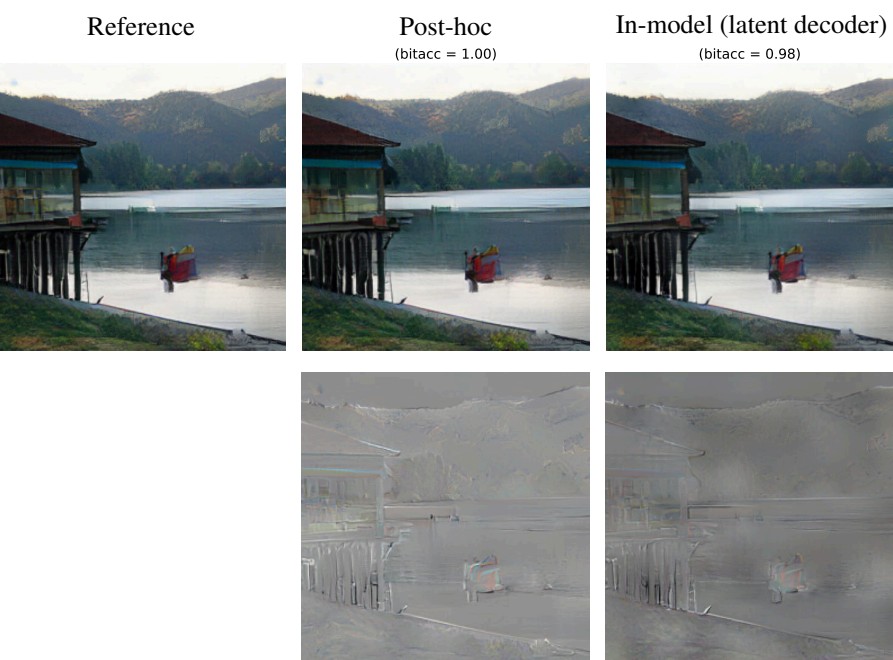

*Figure 8.* Comparison on a validation image of ImageNet between the post-hoc latent watermarker (after the quantization step) and its distillation into the latent decoder of MaskBit. The first column shows the reference image, the second column shows the watermarked image from the post-hoc latent watermarker, and the third column shows the watermarked image from the distilled latent decoder. The bottom row shows the respective difference images.

# J. Additional Experiments on Text-to-Image Task

DISTSEALis a general distillation framework and can be applied to various types of generation tasks. In the main paper, we focus on ImageNet class-to-image as a representative task. In this section, we illustrate with additional experiments on the text-to-image task. We distill the post-hoc watermarking model into the latent encoder of Stable Diffusion XL (SD) (Rombach et al., 2022), which is a popular text-to-image generative model. Additionally, we also distill the latent decoder for comparison. We use Stable Diffusion 1.4 family which relies on U-Net architecture, and use the fine-tuned SD VAE with EMA [2] as the pretrained teacher model. For the post-hoc latent watermarking model, we train the same VideoSeal architecture on the SD latent space ($64 \times 64$), with the same training hyperparameters as for DCAE, except the autoencoder is StableDiffusionVAEv1. The watermarking strength $\epsilon$ is set to 1.5. Here, we did not do an hyperparameter search to find the watermarking strength $\epsilon$ which would yield an imperceptible watermark with the Stable Diffusion autoencoder. We just wanted to show that it is possible to train a post hoc latent watermarking model for Stable Diffusion. For both post-hoc and distilled models, we train on the train split of the MS-COCO dataset (Lin et al., 2014) and evaluate on the validation split. We use the same evaluation protocol as for ImageNet, where we apply various transformations to the generated images and evaluate the bit accuracy under each transformation. Table 17 shows the result of the post-hoc latent teacher, distillation of the generative models, and of the latent decoder with different extractor weights $\lambda_w$. We also provide visual comparisons between the post-hoc latent and distillation results in Figure 9. We can see that the post-hoc latent watermarker achieves 95.75% average bit accuracy across all transformations, while the distilled generative models achieve comparable performance with 94.78% and 96.01% average bit accuracy for the distilled diffusion and latent decoder respectively.

*Table 17.* Results of posthoc-latent and distillation of VideoSeal watermarks to SD VAE on MS-COCO 2017 (5k) across attack families

| Method | Distillation | FID $\downarrow$ | Identity | ValueMetric | Geometric | Compression | Combined | Avg |
|---|---|---|---|---|---|---|---|---|
| Posthoc latent | - | 21.80 | 99.51 | 99.43 | 93.05 | 99.49 | 87.26 | 95.75 |
| Diffusion | - | 21.94 | 99.13 | 98.98 | 92.17 | 99.00 | 84.62 | 94.78 |
| Latent decoder | $\lambda_w = 0.0$ | 22.97 | 99.57 | 99.48 | 93.24 | 99.57 | 88.88 | 96.15 |
| Latent decoder | $\lambda_w = 0.01$ | 23.01 | 99.88 | 99.81 | 93.72 | 99.84 | 89.20 | 96.49 |
| Latent decoder | $\lambda_w = 0.1$ | 22.86 | 99.84 | 99.47 | 92.25 | 98.79 | 89.70 | 96.01 |

---

[2]https://huggingface.co/stabilityai/sd-vae-ft-ema

**COCO prompt: "A cat eating a bird it has caught"**

*Figure 9.* Examples of images generated by the post-hoc latent watermarker, the distilled diffusion model, and the distilled latent decoder for a given COCO prompt. Imperceptibility could be improved by carefully tuning the watermarking strength. We notice that distilling in the diffusion model (second row) actually improves the visual quality compared to the teacher latent post-hoc watermarking model (first row) while achieving comparable bit accuracy.

## K. Analysis of autoregressive DISTSEAL

### K.1. Robustness to latent decoder switching

To address the lack of reverse cycle-consistency in the autoencoder of RAR, (Jovanović et al., 2025) fine-tunes the latent decoder of RAR to enforce cycle-consistency. So, we can test the robustness of our distilled autoregressive model when switching the latent decoder to the fine-tuned cycle-consistent decoder from (Jovanović et al., 2025). We report the results in Table 18, where we observe that the bit accuracy even increases for all the transformations when using the cycle-consistent decoder, showing that our distilled autoregressive model can be robust to changes in the latent decoder.

### K.2. Visualizing token distribution changes at generation time

For the autoregressive distilled model, it is not straightforward to analyze the watermark formation token by token, as the watermark is detected on the full image and not at the token level. Instead, we analyse the *changes* induced by the distillation in the token distribution at generation time. To this end, we generate images where the first half of the tokens are sampled using the pretrained RAR-XL. Then, conditioning on these tokens, we generate the remaining tokens with the distilled model, and we can observe how the token distribution changes for the second half of the tokens. More specifically, for each

*Table 18.* Robustness of the distilled autoregressive RAR-XL when switching the latent decoder to a cycle-consistent decoder from (Jovanović et al., 2025) trained with (row 3) and without augmentations (row 2). We report the FID and bit accuracy under different transformations.

| Latent decoder | FID | IS | Identity | Valuemetric | Geometric | Compression | Combined |
|---|---|---|---|---|---|---|---|
| Original decoder | 3.32 | 288.22 | 94.85 | 93.88 | 85.70 | 93.92 | 77.30 |
| Fine-tuned decoder (no augs) | 4.35 | 274.44 | 96.19 | 95.34 | 87.08 | 95.46 | 81.00 |
| Fine-tuned decoder (with augs) | 5.02 | 265.52 | 96.15 | 95.43 | 87.02 | 95.66 | 81.65 |

generated token by the distilled model, we look whether this token belongs to the top-30 predicted tokens when we compute its logits using the reference RAR-XL transformer (conditioned on the tokens generated so far). This indicates whether the original auto-regressive model would have been likely to generate this token. We visualize these token distribution changes in Figure 10, where we also compare against applying WMAR (Jovanović et al., 2025) for the second half of tokens. We observe that our distilled model leads to significantly more "different" tokens than the KGW sampling (green/red scheme) of WMAR. Indeed, the green tokens of WMAR still correspond to very likely tokens for the original model, while the distillation leads to more significant changes to the token distribution. However, our distilled model still produces images that are visually similar to those of the reference model. Furthermore, the watermark remains detectable (bit accuracy = 0.80) even when only half of the tokens are generated with the distilled model. In Appendix K, we provide more examples of such a comparison.

| Reference | DISTSEAL (bit acc. 0.80) | WMAR | Tokens Diff. (DistSeal) | Tokens Diff. (WMAR) |
|---|---|---|---|---|

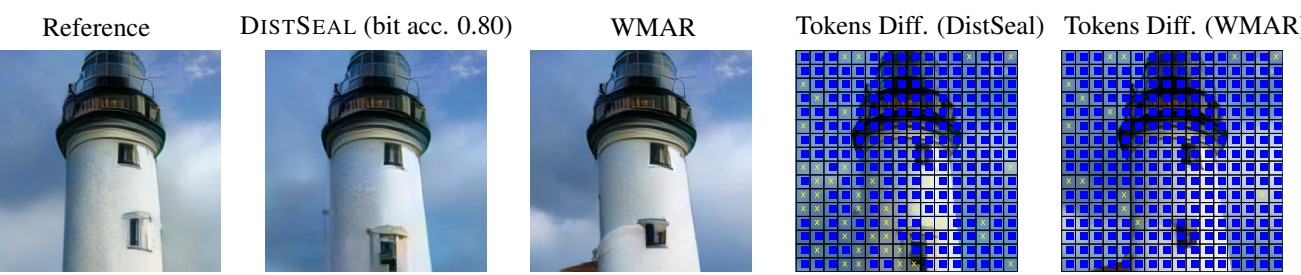

*Figure 10.* Generation of an image (ImageNet class=437) using different autoregressive watermarking models. The first half of tokens is generated using the reference RAR-XL, then the second half is generated by either: reference RAR-XL (left), DISTSEAL distilled transformer (middle), or WMAR on top of the refence RAR-XL (right). The blue "■" or yellow "×" indicates whether the tokens are more or less likely to be generated by the reference RAR-XL (whether the token belongs to the top-30 logits for the reference RAR-XL conditioned on the tokens generated so far). Despite only half of the tokens being generated by the distilled model, the watermark is still detectable (bit accuracy = 0.80).

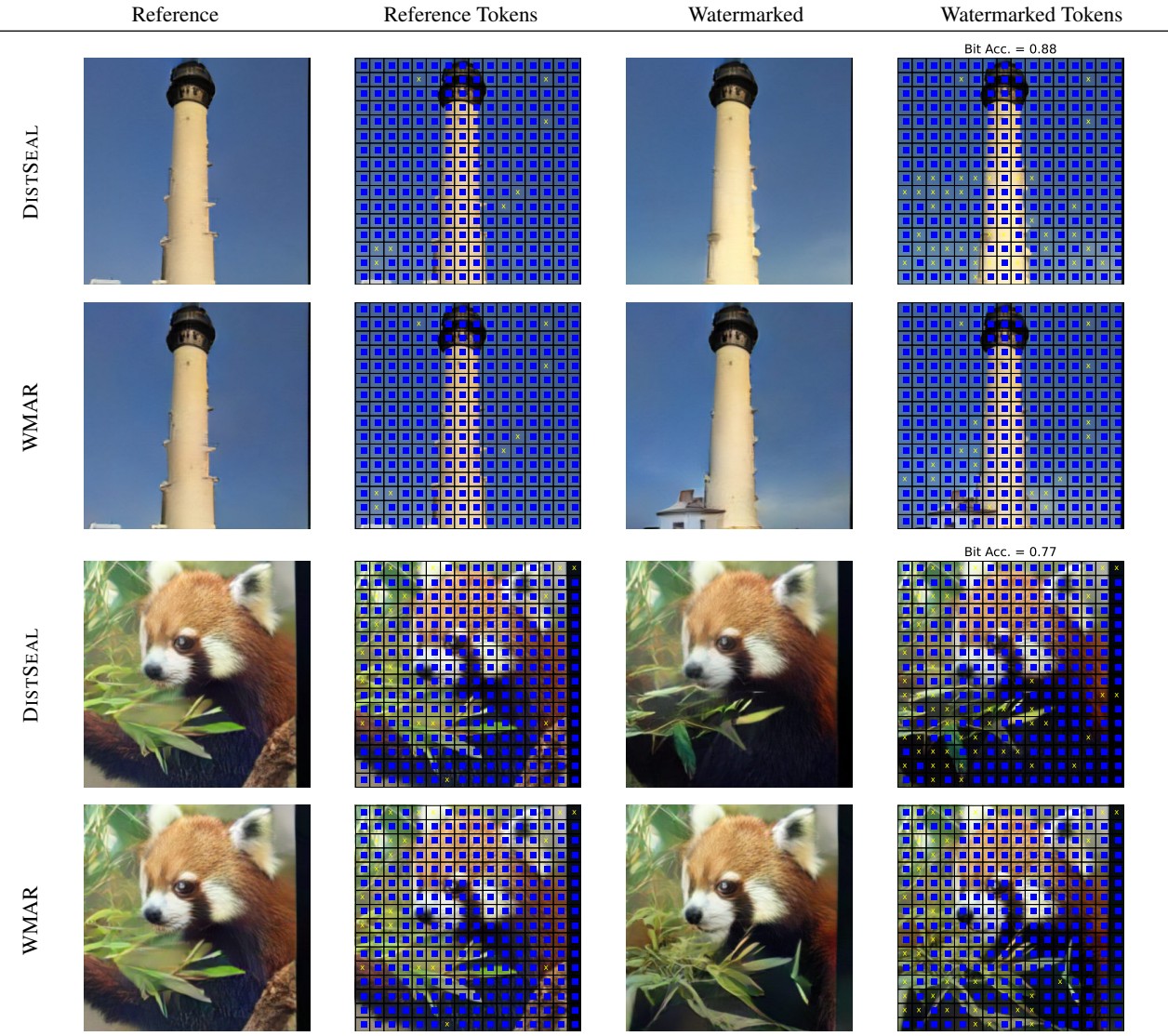

*Figure 11.* Qualitative results of autoregressive models with samples from 3 ImageNet-1k classes. For each class, we generate the first half of tokens using the pretrained RAR-XL model without watermarking, then use the distilled DISTSEALtransformer (top) or WMAR (bottom) to generate the second half of tokens auto-regressively. The token grids show the actual tokens generated by each method, where the blue/yellow tokens indicate whether the tokens are more / less likely to be generated by the reference RAR-XL (whether the token belongs to the top-20 logits for the reference RAR-XL conditioned on the tokens generated so far). For each watermarked image generated by DISTSEAL, we present the bit accuracy detected on the full images.

## L. Effect of the watermarking strength

We provide additional visualizations of watermarked images generated by post-hoc latent watermarkers trained with different watermarking strengths $\epsilon$ in Figure 12, Figure 13 and Figure 14 for DCAE and RAR respectively. As expected, we can see that increasing the watermarking strength generally leads to more visible watermarks. For RAR, we observe that applying the watermark **before** the quantization step (Figure 13) produces more semantic modifications than applying the watermark **after** quantization (Figure 14). We also distill these latent watermarkers trained at different strengths into the diffusion model of DCAE and the autoregressive model of RAR-XL, and report the results in Table 19 and Table 20 respectively. We observe that for all watermarking strengths, the distilled models can closely match the robustness of the teacher models while achieving similar FID scores.

*Table 19.* Distilling post-hoc latent watermarkers trained at at various strengths $\epsilon$ into the diffusion model of DCAE. For each watermark strength, we report the results of the teacher post-hoc watermarker and the distilled model in terms of visual quality (FID) and bit accuracy over transformations.

| Watermark strength | | FID ($\Delta$ ref.) | Identity | Valuemetric | Geometric | Compression | Combined | Avg |
|---|---|---|---|---|---|---|---|---|
| $\epsilon = 0.5$ | Teacher | 11.42 (+0.76) | 99.75 | 98.08 | 91.62 | 99.23 | 84.28 | 95.18 |
| | Distilled | 10.90 (+0.24) | 99.53 | 97.67 | 91.27 | 98.88 | 83.37 | 94.78 |
| $\epsilon = 0.7$ | Teacher | 11.53 (+0.87) | 99.85 | 98.42 | 92.18 | 99.39 | 88.12 | 95.71 |
| | Distilled | 11.02 (+0.36) | 99.76 | 98.20 | 91.99 | 99.22 | 87.46 | 95.49 |
| $\epsilon = 0.9$ | Teacher | 11.54 (+0.88) | 99.90 | 99.03 | 92.62 | 99.53 | 89.45 | 96.24 |
| | Distilled | 11.09 (+0.43) | 99.85 | 98.88 | 92.47 | 99.42 | 88.83 | 96.08 |

*Table 20.* Distilling post-hoc latent watermarkers trained at at various strengths $\epsilon$ into the autoregressive model of RAR-XL. For each watermark strength, we report the results of the teacher post-hoc watermarker and the distilled model in terms of visual quality (FID) and bit accuracy over transformations.

| Watermark strength | | FID ($\Delta$ ref.) | Identity | Valuemetric | Geometric | Compression | Combined | Avg |
|---|---|---|---|---|---|---|---|---|
| $\epsilon = 0.7$ | Teacher | 3.34 (+0.19) | 87.90 | 86.13 | 79.02 | 86.01 | 59.94 | 79.80 |
| | Distilled | 3.34 (+0.19) | 86.97 | 85.59 | 78.79 | 85.50 | 62.54 | 79.88 |
| $\epsilon = 1.0$ | Teacher | 3.42 (+0.28) | 93.30 | 91.85 | 84.12 | 91.85 | 68.88 | 86.00 |
| | Distilled | 3.36 (+0.22) | 92.41 | 91.22 | 83.79 | 91.19 | 71.21 | 85.96 |
| $\epsilon = 1.5$ | Teacher | 3.56 (+0.42) | 96.92 | 95.91 | 87.20 | 95.98 | 77.00 | 90.60 |
| | Distilled | 3.31 (+0.17) | 94.85 | 93.88 | 85.70 | 93.92 | 77.30 | 89.13 |

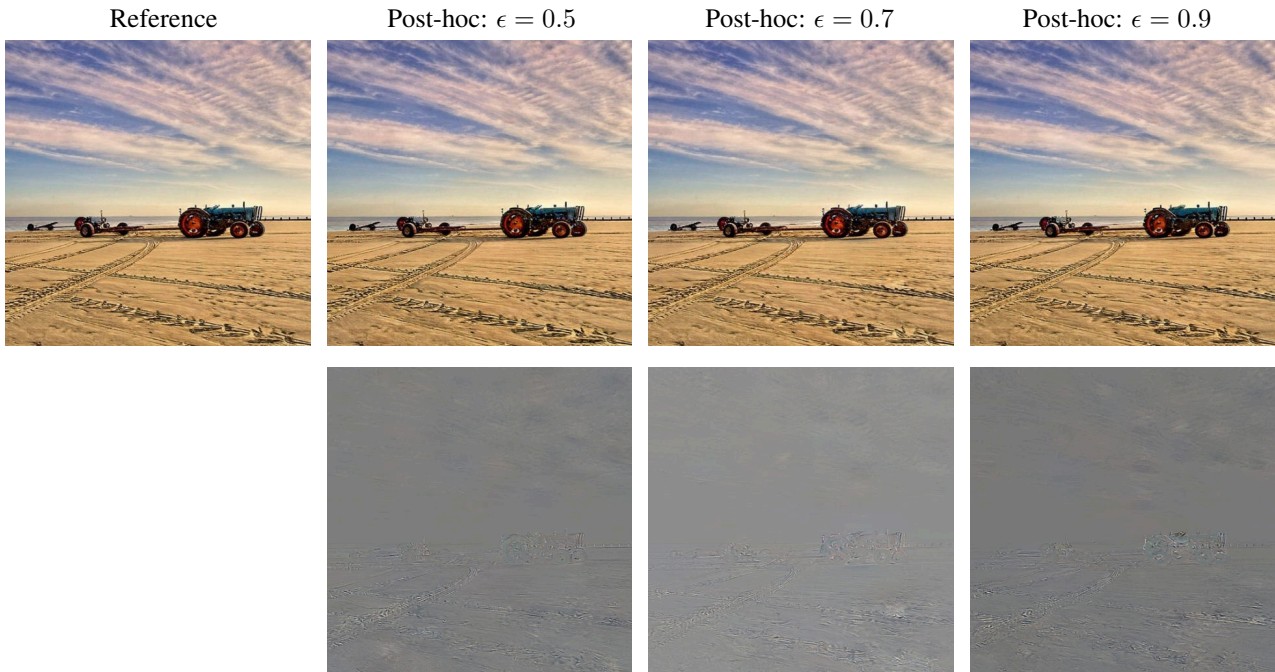

*Figure 12.* Visualizing three DCAE post-hoc latent watermarkers on a validation image of ImageNet. Each latent watermarker was trained with different watermarking strengths $\epsilon$. The first row shows the watermarked images generated by each post-hoc model, and the second row shows the respective diff images between the reference and watermarked images.

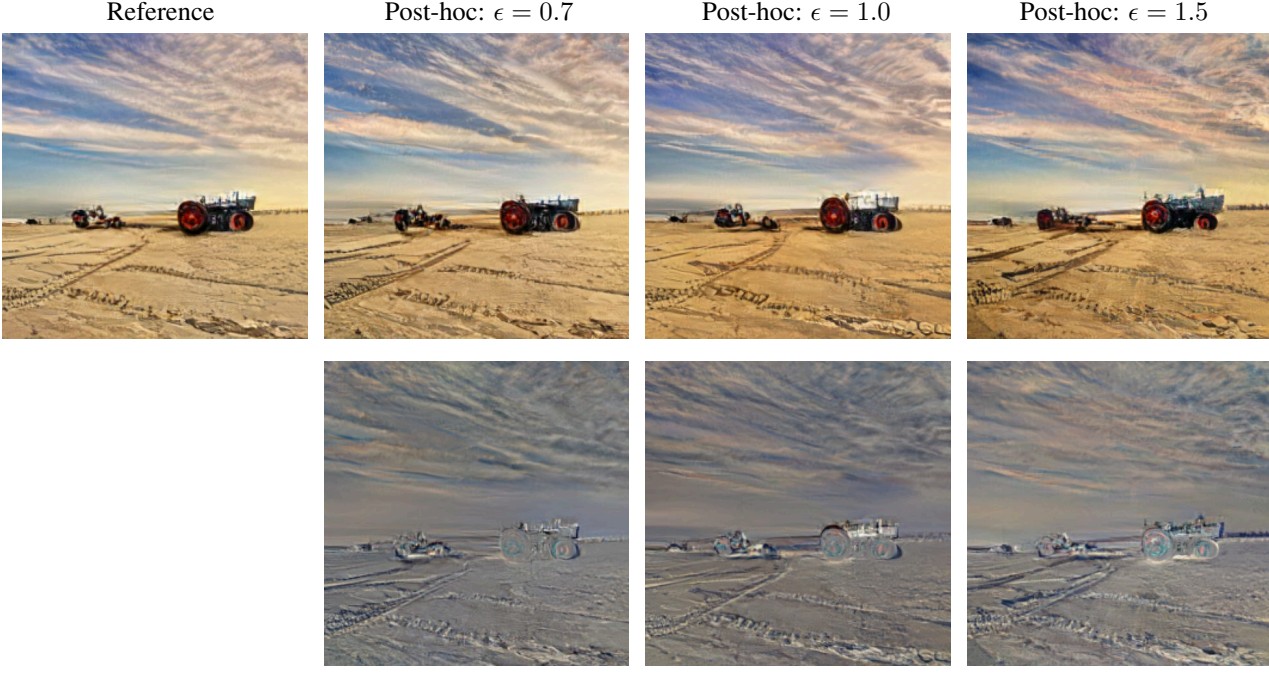

*Figure 13.* Visualizing three RAR post-hoc latent watermarkers (applied **before** the quantization step) on a validation image of ImageNet. Each latent watermarker was trained with different watermarking strengths $\epsilon$. The first row shows the watermarked images generated by each post-hoc model, and the second row shows the respective diff images between the reference and watermarked images. We can see that the watermarks produce semantic modifications.

Reference Post-hoc: $\epsilon = 0.5$ Post-hoc: $\epsilon = 0.75$ Post-hoc: $\epsilon = 1.0$

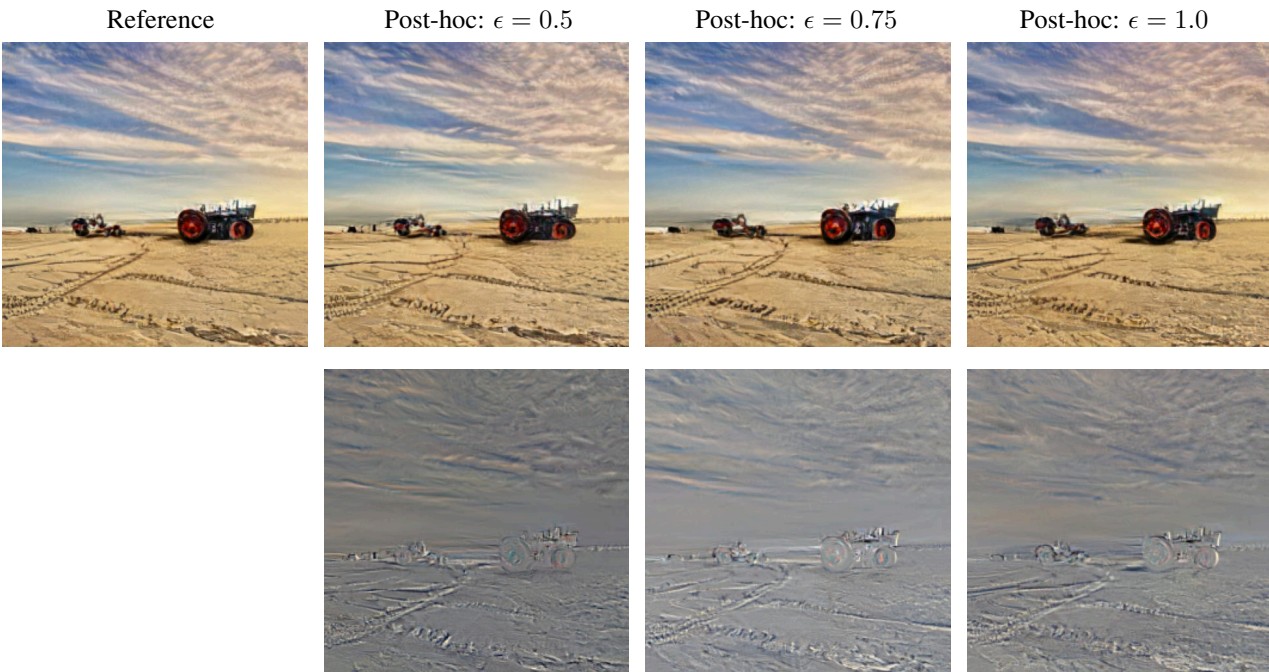

*Figure 14.* Visualizing three RAR post-hoc latent watermarkers (applied **after** the quantization step) on a validation image of ImageNet. Each latent watermarker was trained with different watermarking strengths $\epsilon$. The first row shows the watermarked images generated by each post-hoc model, and the second row shows the respective diff images between the reference and watermarked images. We can see that the watermarks produce less semantic modifications than in Figure 13 where the watermarks are applied before quantization.

# M. More visualizations

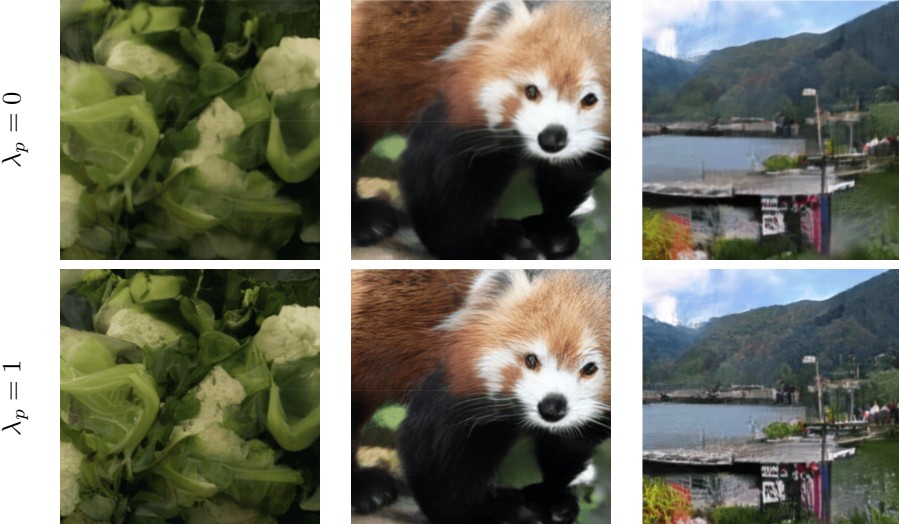

*Figure 15.* Influence of the LPIPS perceptual loss weight $\lambda_p$ when distilling into the latent decoder of RAR. We observe that not using the perceptual loss (top row with $\lambda_p = 0$) results in blurrier images. Interestingly, the latent decoder of DCAE does not suffer from blurry images when not using LPIPS in the reconstruction loss.

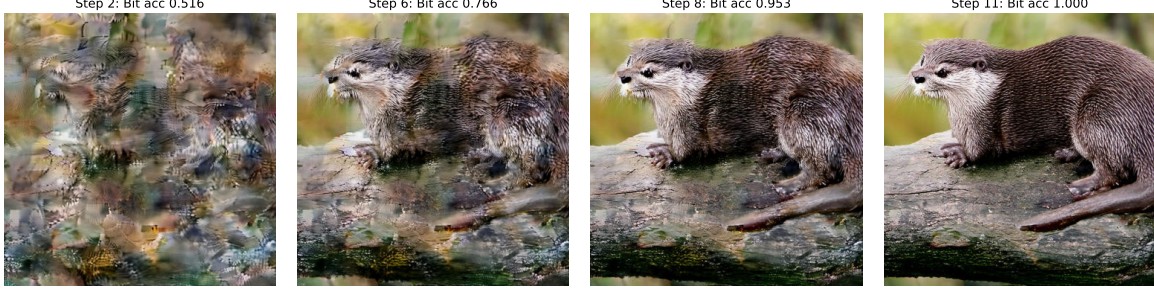

*Figure 16.* Visualizing the image quality evolution over diffusion steps and reporting the bit accuracy at each step. We can see that as the diffusion process proceeds, the image quality improves and the watermark bit accuracy increases accordingly.

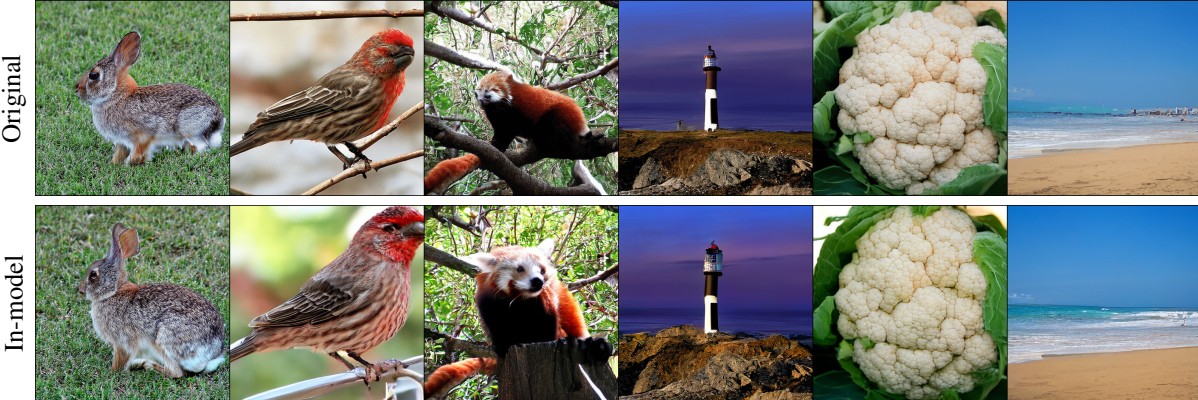

*Figure 17.* We show several generation examples for the same seed for the DCAE diffusion model before and after distillation of the latent watermarking model. The first row shows original images generated by the pretrained DCAE diffusion model, and the second row shows images generated by the distilled diffusion model.

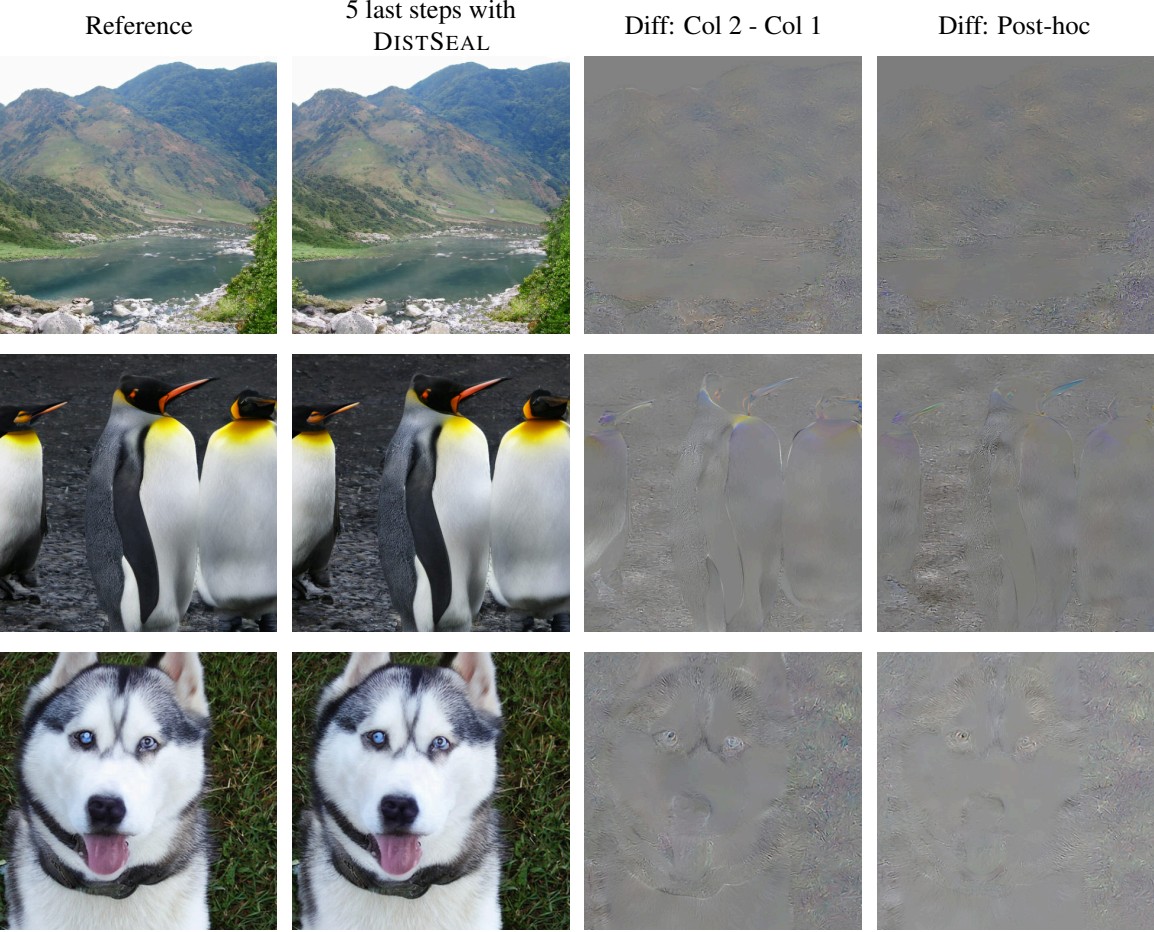

*Figure 18.* Visualizing the watermark for the distilled DCAE diffusion model. The first column shows images generated by the pretrained DCAE diffusion model, and the second column shows images where the last 5 steps are generated by the distilled model. In the third column, we show the difference images between the first two columns and compare with applying the post-hoc latent watermarking model to the latents of the first column.

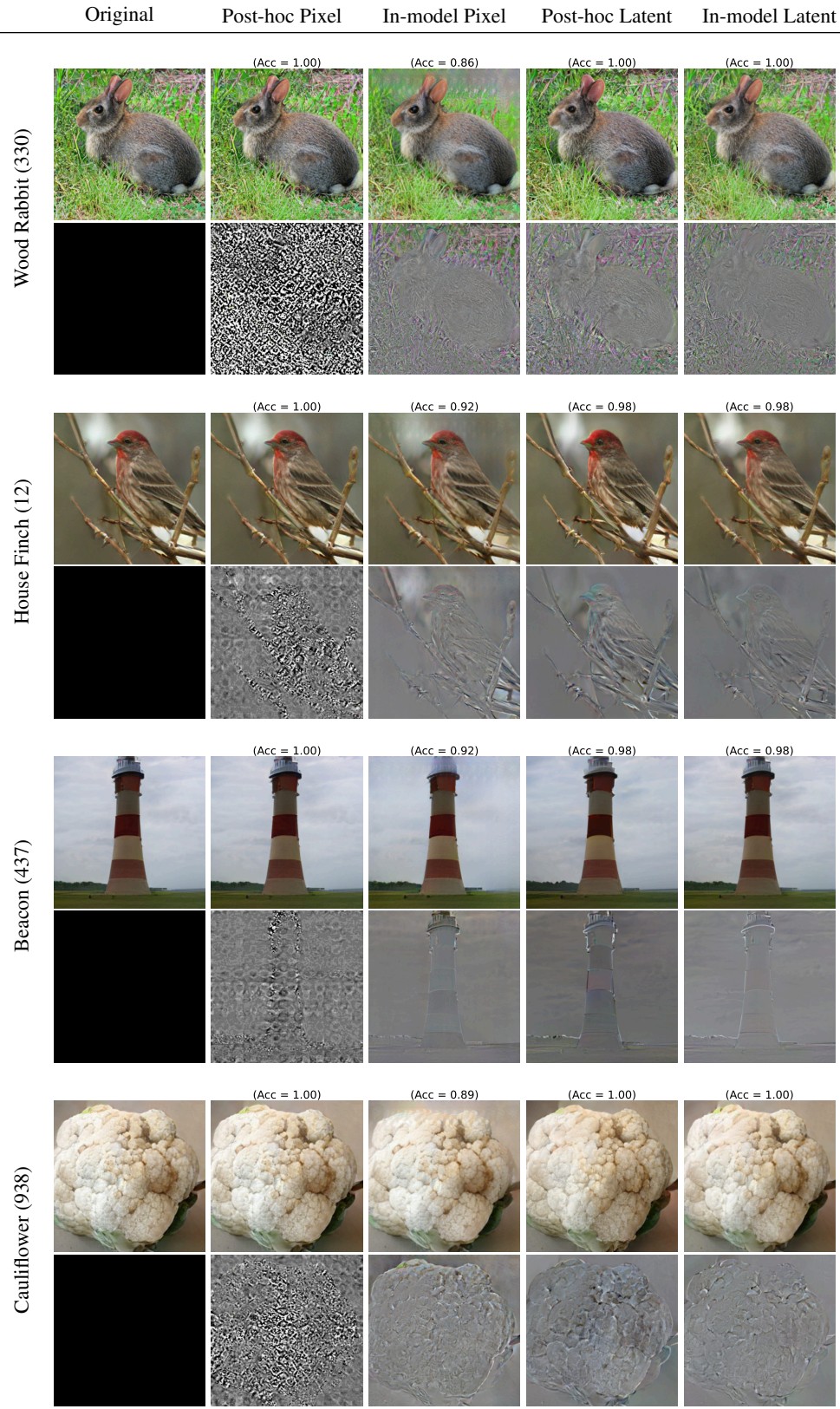

*Figure 19.* Comparing RAR-XL post-hoc watermarkers and their distilled latent decoders on generated images. The first column shows the original images generated by the pretrained RAR-XL model, the second and fourth columns show watermarked images from post-hoc watermarkers applied at the pixel level and latent level respectively, and the third and fifth columns show watermarked images from the respective distilled latent decoders. The bottom two rows show the respective difference images between the original and watermarked images for each method.

