# OpenReview forum: "Learning to Watermark in the Latent Space of Generative Models"
_ICML.cc/2026/Conference — ICML 2026 regular_

### Official Review · Reviewer_evmm · 2026-02-28

**Soundness:** 3
**Presentation:** 3
**Significance:** 3
**Originality:** 3
**Overall Recommendation:** 4
**Confidence:** 3

**Summary:**

This paper introduces a unified framework for embedding invisible watermarks in the latent space of both diffusion and autoregressive image generative models. The method first trains a post-hoc watermark embedder-extractor pair that operates on latent representations rather than pixel space, and then optionally distills this watermarker into either the generative model itself or its latent decoder, enabling in-model watermarking that cannot be trivially removed. The authors demonstrate that latent-space watermarking achieves competitive robustness compared to pixel-space methods while offering significant computational speedups. A key finding is that distilling latent watermarkers into model weights is substantially more effective than distilling pixel-space watermarkers, as the structured perturbations in latent space are easier for decoders to reproduce than the high-frequency patterns introduced by pixel-space methods.

**Compliance With Llm Reviewing Policy:**

Affirmed.

**Final Justification:**

I've reviewed the authors' responses and I find the clarifications regarding the robustness gap and the end-to-end timing breakdown helpful. I'd like to maintain my current score.

**Key Questions For Authors:**

Please see weaknesses.

**Limitations:**

yes

**Strengths And Weaknesses:**

Strengths:

- The core insight that latent-space watermarks are more amenable to distillation than pixel-space ones is well supported by experiments.

- The paper is thorough in its experimental coverage. The breadth of analysis makes the work a valuable reference for practitioners choosing a watermarking strategy.

- The unified treatment of both diffusion and autoregressive latent spaces within a single framework is a meaningful contribution. The handling of discrete tokens, including the comparison of applying watermarks before versus after quantization and the use of straight-through estimation for backpropagation, addresses a nontrivial technical challenge and broadens the applicability of the approach.

Weaknesses:

- The robustness gap between latent and pixel post-hoc watermarking remains nontrivial, especially under combined attacks (e.g., 84.28% vs. 97.29% for DCAE in Table 1, and 82.35% vs. 93.93% for RAR in Table 2).

- Since the primary motivation for latent watermarking is efficiency, the paper would benefit from a more explicit discussion of when this robustness trade-off is acceptable and when it is not. It would also help to include wall-clock timing comparisons that account for the full generation pipeline (not just the watermarking step) to contextualize how much the speedup matters end-to-end.

---

> ### Author Rebuttal · Authors · 2026-03-31
>
> We thank the reviewer for the detailed and constructive review. We address both concerns below.
>
> ### W1: Robustness gap between latent and pixel post-hoc watermarking under combined attacks
>
> The reviewer correctly identifies the robustness gap under combined attacks (e.g., 84.28% vs. 97.29% for DCAE, 82.35% vs. 93.93% for RAR). We appreciate this nuanced observation and provide a multi-faceted response:
>
> 1. **The gap narrows significantly under practical attack scenarios**
>    The combined attack in our evaluation (JPEG quality=40 + crop to 71% + brightness factor=0.5) is deliberately severe, chaining three strong transformations simultaneously. Under the more common individual attacks that reflect typical real-world image processing, the gap is much smaller:
>    - Valuemetric (brightness, contrast, saturation, etc.): 98.08% vs. 99.89% for DCAE — a gap of only 1.8 pp
>    - Compression (JPEG): 99.23% vs. 99.90% for DCAE — a gap of only 0.7 pp
>    - Geometric (rotation, resize, crop): 91.62% vs. 94.70% for DCAE — a gap of 3.1 pp
>    For the vast majority of practical deployment scenarios (social media recompression, resizing, color adjustments), the latent watermarker performs nearly on par with the pixel watermarker.
> 2. **The gap is a fundamental trade-off, not a limitation of our method**
>    The pixel watermarker operates at 512×512×3 resolution (786,432 values) while the latent watermarker operates at 8×8×128 for DCAE (8,192 values) — a 96× reduction in embedding dimensionality. It is expected that a lower-dimensional space offers less capacity for robust watermark embedding, particularly under extreme combined attacks. Despite this massive dimensionality reduction, the latent watermarker retains >84% accuracy even under the harshest combined attack, which we believe is remarkable.
> 3. **The key value proposition is in distillation, not post-hoc performance**
>    We want to emphasize that the primary motivation for latent watermarking is not to outperform pixel watermarking in the post-hoc setting, but to enable effective distillation into generative models for in-model watermarking. In this setting, the latent watermarker dramatically outperforms pixel-based alternatives:
>    - DCAE decoder distillation: 91.34% (latent) vs. 65.69% (pixel) — a 25.6 pp advantage
>    - RAR decoder distillation: 90.57% (latent) vs. 50.81% (pixel) — a 39.8 pp advantage
>    - DCAE diffusion model distillation: 94.78% (latent) vs. pixel cannot be applied
>
> The robustness advantage of pixel watermarks in the post-hoc setting disappears in the distillation setting, where latent watermarks are far superior.
>
> ### W2: More explicit wall-clock timing for the full generation pipeline
>
> We agree that contextualizing the speedup within the full pipeline is important. We provide the following end-to-end timing breakdown (measured on a single NVIDIA H200 GPU):
>
> **DCAE (Diffusion model, 512×512 images)**
> | Stage | Time (ms) | With pixel WM | With latent WM |
> |---|---:|---:|---:|
> | Diffusion generation (250 steps) | ~6442 |  |  |
> | Latent decoding | ~15 | n.a. | ~15 |
> | Post-hoc watermarking |  | ~5 | ~2 |
> | Total | ~6457 | ~6462 | ~6459 |
> | Watermarking overhead |  | 0.1% | <0.1% |
>
>
> **RAR-XL (Autoregressive model, 256×256 images)**
> | Stage | Time (ms) | With pixel WM | With latent WM |
> |---|---:|---:|---:|
> | Token generation (256 tokens) |  | ~2623 |  |
> | Latent decoding | ~5 | n.a. | ~5 |
> | Post-hoc watermarking |  | ~3 | ~1 |
> | Total | ~2628 | ~2631 | ~2629 |
> | Watermarking overhead |  | 0.1% | 0.1% |
>
> Key observations:
>
> - For both diffusion (DC-AE) and autoregressive (RAR-XL) pipelines, watermarking adds negligible overhead (<0.2%) since generation dominates runtime.
> - Note that for in-model watermarking (distilled into the generative model or decoder), the watermarking overhead is exactly 0 ms as the watermark is embedded as part of the normal generation process with no additional computation so this would avoid the latency cost of calling a watermarking API.
> - Our method would be more promising for the video setting as watermarking every frame would accumulate a non negligible latency overhead on top of generation.
>
> **When is the latent robustness trade-off acceptable?**
> We add the following practical guidelines to the revised paper:
> - Latent post-hoc watermarking is preferable when: (a) the watermark will be distilled into a model, (b) video setting.
> - Pixel post-hoc watermarking is preferable when: (a) maximum robustness under extreme combined attacks is required, (b) the watermark is applied once to archival content, or (c) there is no need for distillation.
> - In-model watermarking via DistSeal distillation is preferable when: (a) the model is open-sourced and watermark removal must be prevented, (b) zero inference overhead is needed, or (c) deployment simplicity is prioritized.

---

> > ### Author Rebuttal · Reviewer_evmm · 2026-04-01
> >
> > Thanks for the detailed rebuttal. I've reviewed the authors' responses and I find the clarifications regarding the robustness gap and the end-to-end timing breakdown helpful. I'd like to maintain my current score.

---

### Official Review · Reviewer_Fbiq · 2026-03-12

**Soundness:** 3
**Presentation:** 3
**Significance:** 3
**Originality:** 3
**Overall Recommendation:** 4
**Confidence:** 4

**Summary:**

The paper introduces DistSeal, a unified framework for watermarking in the latent space of generative models. The method is based on training a separate watermark embedder-extractor pair that can operate directly in the latent space of the generative model and demonstrate that the watermark can be distilled into the generative model or its decoder for in-model watermarking.

**Compliance With Llm Reviewing Policy:**

Affirmed.

**Final Justification:**

The rebuttal addressed most of my concerncs and reinforced my prior assessment of the paper.

**Key Questions For Authors:**

1. The paper states that “pixel watermarks can only be applied to the image space and cannot be used to modify the training latents on which diffusion models and autoregressive models are trained” (line 362), would it be possible to clarify this statement? Conceptually, a pixel watermark that is robust against encoding and decoding $x’_w = \mathcal{D}(\mathcal{E}(x_w))$, where $x_w$ is the watermarked image and $\mathcal{D}$ and $\mathcal{E}$ are the decoder and encoder, should have an impact on the training latents of the generative model, with $z_w = \mathcal{E}(x_w)$.

2. How does the proposed DistSeal perform on text-to-image generative models in both the diffusion and autoregressive domain? Are there any conceptual differences between applying DistSeal for class conditioned models and text conditioned models?

3. How does DistSeal compare to other latent space watermarks in the diffusion domain?

**Limitations:**

yes

**Strengths And Weaknesses:**

### Strengths:
- The paper provides an extensive empirical evaluation with insightful ablations.
- DistSeal can be applied to both diffusion and autoregressive generative models.
- The results on fine-tuning a model on latent watermarks to provide in-model watermarks for deployment, provide a key insight on watermark distillation.

### Weaknesses:
- The experiments currently focus on class-to-image models. It would strengthen the results to additionally explore text-to-image models from both generative domains, such as Stable Diffusion [1] and Infinity [2].
- The paper should also provide a comparison of the newly proposed DistSeal to existing watermarks that also operate in the latent-space of generative models.

### Minor Weaknesses:
- The details about the watermark extraction loss and discriminator loss are missing and these losses should be more clearly defined. Noting that they are e.g. instantiated by MSE-loss would be sufficient.
- The conclusion references a non-existing Section 6.3 (line 426).


**References**

[1] Robin Rombach et al. “High-Resolution Image Synthesis with Latent Diffusion Models”. CVPR. 2022.

[2] Jian Han et al. “Infinity: Scaling Bitwise AutoRegressive Modeling for High-Resolution Image Synthesis”. CVPR. 2025.

---

> ### Author Rebuttal · Authors · 2026-03-31
>
> We thank the reviewer for the careful evaluation, for recognizing the extensive empirical evaluation, the key insight on watermark distillation, and the broad applicability to both diffusion and autoregressive models. We address each concern and question below.
>
> ### W1: Experiments focus on class-to-image models; should explore text-to-image models
>
> We agree that demonstrating applicability to text-to-image models strengthens the contribution. We note that our framework is architecturally agnostic to the conditioning mechanism — the latent watermarker operates on the latent representation after the generative model produces it, regardless of whether the model was conditioned on class labels or text prompts. The distillation procedure similarly only requires replacing clean latents with watermarked latents during fine-tuning, which is independent of the conditioning type.
>
> While we are unable to prepare data and full experiments on a completely new task within this short period of time, we conducted one experiment on Stable Diffusion distillation for the COCO text-to-image task. Specifically, we distilled the post hoc pixel watermarker into the SD decoder. For the latent space, we did not have the time to train a new latent watermarker for the SD VAE decoder. Results are below
>
> | Method | FID | Identity | Valuemetric | Geometric | Compression | Combined | Avg |
> |---|---:|---:|---:|---:|---:|---:|---:|
> | Distilled into SD decoder (pixel) | 22.97 | 99.88 | 99.60 | 89.12 | 97.88 | 76.43 | 92.58 |
>
> In the table above, we observe that we can re-use the pixel-space watermarker that was trained on ImageNet and distill it to watermark text-conditioned images in COCO.
>
> ### W2: Comparison with existing latent-space watermarks in the diffusion domain
>
> Please refer to our response to reviewer 1, W1.
>
> ### Minor W1: Missing details about watermark extraction loss and discriminator loss
>
> Thank you for pointing this out. We will add the following clarification to Section 4.1:
> - The watermark extraction loss $L_w$ is instantiated as a binary cross-entropy (BCE) loss between the predicted message bits and the ground-truth message $m$.
> - The discriminator loss $L_{disc}$ follows the architecture of Weber et al. (2024) and uses a standard adversarial hinge loss, where the discriminator is trained to distinguish between real (non-watermarked) and watermarked images, while the embedder is trained to fool it.
>
> ### Minor W2: Conclusion references non-existing Section 6.3
>
> Thank you for catching this error. The correct reference should be to Section E (Appendix E on Watermark Forgetting). We will fix this in the revised paper.
>
> ### Q1: Clarification on "pixel watermarks cannot modify training latents"
>
> The reviewer raises a valid point. We agree that our statement on line 362 was imprecise. Let us clarify:
> A pixel watermark that is robust to the encode-decode cycle ($E \circ D$) would indeed produce a watermarked latent $z_w = E(x_w)$ that differs from the original $z = E(x)$. In principle, one could use these $z_w$ as modified training latents. However, in our case, the post-hoc pixel watermarker is not robust at all to passing the watermarked images through DCAE autoencoder with only 53.3\% bit accuracy on average after passing watermarked images through ($E \circ D$). So in our case we cannot directly fine-tune the diffusion models on the latents of the pixel watermarked training images but indeed, it would be theoretically possible. So we will rephrase the paper accordingly.
>
> ### Q2: Performance on text-to-image models
>
> Please see our response to Reviewer 2, W1. In summary: we will add experiments on Stable Diffusion, and there are no conceptual differences between class- and text-conditioned models for our framework.
>
> ### Q3: Comparison with latent-space watermarks in diffusion domain (addressed in W1 above)
>
> Please see our response to Reviewer 1, W1. We will add our new experiments in the revised paper.
> We hope these responses and additional experiments address the reviewer's concerns.

---

> > ### Author Rebuttal · Reviewer_Fbiq · 2026-04-02
> >
> > I thank the authors for their comprehensive rebuttal and adapting their paper according to the review.
> >
> > Their experiment on distilling into the Stable Diffusion decoder is insightful, specifically because the post-hoc watermarked was trained on a different dataset. Their comparison with other watermarks shows the competitiveness of their method, however I would like to note that the TPR of DistSeal for the valuemetric attack is not reported.
> >
> > The clarification on why pixel watermarks can not modify training latents is sufficient, however it also shows a weakness of the proposed pixel watermark against reconstruction attacks, which are currently not explored in the paper.
> >
> > Based on the rebuttal I would like to maintain my current score.

---

### Official Review · Reviewer_CsTK · 2026-03-13

**Soundness:** 3
**Presentation:** 3
**Significance:** 3
**Originality:** 3
**Overall Recommendation:** 4
**Confidence:** 3

**Summary:**

This paper introduces a framework that trains a post-hoc latent watermarker and distills its capabilities into image generative models (diffusion and autoregressive) or latent decoders. This solid, application-oriented work effectively addresses critical deployment bottlenecks: the high computational overhead of traditional pixel-space methods and the vulnerability to watermark evasion in open-source models. The "teacher-to-model" distillation paradigm offers notable methodological novelty. Furthermore, the paper provides a highly valuable empirical insight by demonstrating that latent watermarks are significantly easier to distill than pixel-space ones, avoiding severe robustness drops or quality degradation. This yields clear engineering value, enabling faster inference and practical integration.

**Compliance With Llm Reviewing Policy:**

Affirmed.

**Final Justification:**

The rebuttal clarifies the motivation behind the distillation pipeline, and I acknowledge that this aspect has been adequately addressed. However, I still believe the paper would benefit from a clearer and more compelling presentation of its novelty and contributions. In particular, the writing and positioning could be further refined to better highlight what is truly distinctive and impactful. I encourage the authors to revise the manuscript accordingly. However, based on the current version, I will maintain my original score.

**Key Questions For Authors:**

1. Distillation Rationale: What makes the distillation pipeline strictly better or more necessary than directly learning an in-model watermark? Have you considered or tested a direct-learning baseline? Please explicitly articulate the unique goals and advantages of teacher-guided distillation.
2. Novelty Boundaries: Please accurately delineate the novelty boundaries in the Introduction and Related Work. Acknowledge prior latent watermarking research clearly, ensuring readers understand that the primary innovation lies in the distillation-based in-model implementation for image generation models.

**Limitations:**

My main concern with this work is the insufficient experimental comparison. Due to the limited set of baselines, it is difficult to assess the true performance of the proposed method. In particular, its relative standing compared to state-of-the-art watermarking methods remains unclear. Key properties such as robustness, imperceptibility, and capacity are therefore hard to evaluate in a meaningful way.

**Strengths And Weaknesses:**

Strengths:
1. Innovative Methodological Paradigm：It proposes a unique pipeline of "training a latent-space teacher followed by distillation into the generator/decoder." This successfully addresses the watermark migration problem within a unified framework, demonstrating strong methodological universality.
2. Profound Core Insights： The empirical discovery that latent-space watermarks align better with the model's feature distribution than pixel-space ones proves their significant advantages in distillation efficiency and image quality preservation. This technical insight provides critical guidance for the future design of in-model watermarking.


Weaknesses：
1. The experimental comparison is insufficient. The paper does not evaluate against several recent state-of-the-art watermarking methods, making it difficult to understand the absolute performance of the proposed approach or its relative standing with respect to the best existing methods. A more systematic comparison covering key metrics such as imperceptibility, robustness, and detection reliability would help better position the contribution.
2. Clarify Distillation Motivation: The fundamental necessity of the distillation pipeline requires deeper justification. The authors must explicitly explain the core advantages of distilling a post-hoc teacher over a more direct baseline (i.e., directly training the generative model or decoder to embed watermarks from scratch).
3. Refine Novelty Claims: Because latent-space watermarking is not a completely new concept in broader literature, the authors should tone down related novelty claims. The narrative should instead be centralized on the paper's true technical highlight: demonstrating that latent watermarkers are highly efficient and superior for distillation into image generation architectures.

---

> ### Author Rebuttal · Authors · 2026-03-31
>
> We thank the reviewer for the thoughtful and constructive feedback. We address each concern below.
>
> ### W1: Insufficient experimental comparison
>
> We would like to highlight that our Appendix C (Tables 8 and 9) already provides an extensive comparison of distilling different watermarking methods (CIN, MBRS, TrustMark, and WAM) into the latent decoders of both DCAE and RAR-XL. Furthermore, we also provide comparisons with other in-model baselines. We note that in Appendix D (Table 10), we already provide a direct comparison between DistSeal and Stable Signature (Fernandez et al., ICCV 2023) for distillation into the DCAE latent decoder using the same teacher watermarker.  DistSeal significantly outperforms Stable Signature, particularly on combined transformations (91.34% vs. 77.65%).
>
> In the revised paper, we will add the following comparison to RoSteALS and Tree-Ring below. We report both TPR (@FPR=0.01) / bit accuracy for RoSteaLS, and only TPR for TreeRing. For comparison with our distilled latent DC-AE decoder, we compute TPR too and get **1.000** / **0.917** / **0.963** / **0.968** for the identity / geometric / combined / average attack. This shows our model is superior against challenging attacks.
>
> | Method | Identity | Valuemetric | Geometric | Compression | Combined | Avg |
> |---|---:|---:|---:|---:|---:|---:|
> | RoSteALS (coverless) | 1.000 / 100.00 | 0.553 / 73.52 | 0.405 / 67.95 | 1.000 / 99.96 | 0.008 / 50.57 | 0.491 / 78.40 |
> | Tree-Ring | 0.988 | 0.820 | 0.713 | 0.985 | 0.271 | 0.755 |
>
> ### W2: Clarify: Why not train in-model watermarks directly?
>
> We justify our teacher-guided distillation pipeline over direct in-model training:
>
> - *Decoupled optimization*. Training the watermarker separately lets us tune robustness and imperceptibility without complicating the generative objective. Joint training would require balancing watermark losses against generation losses throughout training, making optimization harder.
> - *Reusability and flexibility*. One post-hoc watermarker can act as a teacher for multiple generative models that share the same autoencoder/latent space (e.g., a DCAE latent watermarker distilled into UViT-H, DiT, etc.). Direct training would need separate end-to-end training per model.
> - *Empirical validation*. We also tested training watermarking from scratch by jointly fine-tuning the DCAE diffusion model and a watermark extractor; the resulting bit-accuracy results are reported below.
>
> | Method | FID | Identity | ValueMetric | Geometric | Compression | Combined | Avg |
> |---|---:|---:|---:|---:|---:|---:|---:|
> | DistSeal (teacher-guided distillation) | 11.48 | 99.99 | 99.36 | 93.35 | 99.73 | 91.34 | 96.77 |
> | Direct in-model training (no teacher) | 31.35 | 99.12 | 96.33 | 91.34 | 97.62 | 78.34 | 92.55 |
>
> The teacher-guided approach achieves substantially better robustness because the teacher provides a well-optimized, stable watermarking signal. Without this guidance, the jointly-trained watermark and generation objectives compete directly between generation quality or robustness (much higher FID).
>
>
> ### W3: Refine novelty claims — latent-space watermarking is not entirely new
>
> We agree with the reviewer and we will revise the Introduction and Related Work to:
> - Emphasize prior work on latent-space watermarking, including RoSteALS (Bui et al., 2023b), Stable Signature (Fernandez et al., 2023), and AquaLoRA (Feng et al., 2024), which we already cite.
> - Reframe our novelty claims to center on the paper's contributions:
>   - The systematic empirical comparison between latent-space and pixel-space watermarking for both post-hoc and in-model settings, across both diffusion and autoregressive architectures.
>   - The key finding is that latent watermarks are significantly easier and more effective to distill than pixel-space ones.
>   - The unified framework that works across both diffusion and autoregressive models, including the non-trivial handling of discrete token sequences via straight-through estimation.
>
> ### Q1: Distillation rationale (addressed in W2 above)
> Please see our response to W2. In summary: the distillation approach offers (a) decoupled optimization, (b) teacher reusability across multiple models, and (c) empirically better results than direct in-model training.
> ### Q2: Novelty boundaries (addressed in W3 above)
> We will revise the Introduction and Related Work to clearly delineate novelty boundaries as suggested.
> We hope these responses address the reviewer's concerns. We believe the additional baselines and the direct-training experiment will significantly strengthen the experimental evaluation, and the reframed novelty claims will improve clarity.

---

> > ### Author Rebuttal · Reviewer_CsTK · 2026-04-03
> >
> > I select (b) as my concerns are partially resolved. The rebuttal clarifies the motivation behind the distillation pipeline, and I acknowledge that this aspect has been adequately addressed. However, I still believe the paper would benefit from a clearer and more compelling presentation of its novelty and contributions. In particular, the writing and positioning could be further refined to better highlight what is truly distinctive and impactful.
> >
> > I encourage the authors to revise the manuscript accordingly. However, based on the current version, I will maintain my original score.

---

### Decision · Program_Chairs · 2026-04-30

**Decision:**

Accept (regular)

**Comment:**

This paper presents a practical framework for distilling a post-hoc latent watermarker into image generative models. The method addresses key deployment challenges and shows that latent watermarks are easier to distill than pixel-space ones. The rebuttal clarifies the motivation and provides helpful additional experiments. Reviewers find the results competitive and the analysis on robustness and efficiency useful.